# Cebp1 and Cebpβ transcriptional axis controls eosinophilopoiesis in zebrafish

Gaofei Li[1,2,9], Yicong Sun[2,9], Immanuel Kwok ®[3], Liting Yang[4], Wanying Wen[2], Peixian Huang[2], Mei Wu[2], Jing Li[2], Zhibin Huang ®[2], Zhaoyuan Liu ®[5], Shuai He ®[6], Wan Peng[6], Jin-Xin Bei ®[6], Florent Ginhoux ®[3,5], Lai Guan Ng ®[3,7,8] ✉ & Yiyue Zhang ®[1,2] ✉

Eosinophils are a group of granulocytes well known for their capacity to protect the host from parasites and regulate immune function. Diverse biological roles for eosinophils have been increasingly identified, but the developmental pattern and regulation of the eosinophil lineage remain largely unknown. Herein, we utilize the zebrafish model to analyze eosinophilic cell differentiation, distribution, and regulation. By identifying *eslec* as an eosinophil lineage-specific marker, we establish a *Tg(eslec:eGFP)* reporter line, which specifically labeled cells of the eosinophil lineage from early life through adulthood. Spatial-temporal analysis of *eslec*+ cells demonstrates their organ distribution from larval stage to adulthood. By single-cell RNA-Seq analysis, we decipher the eosinophil lineage cells from lineage-committed progenitors to mature eosinophils. Through further genetic analysis, we demonstrate the role of Cebp1 in balancing neutrophil and eosinophil lineages, and a Cebp1-Cebpβ transcriptional axis that regulates the commitment and differentiation of the eosinophil lineage. Cross-species functional comparisons reveals that zebrafish Cebp1 is the functional orthologue of human C/EBPε[P27] in suppressing eosinophilopoiesis. Our study characterizes eosinophil development in multiple dimensions including spatial-temporal patterns, expression profiles, and genetic regulators, providing for a better understanding of eosinophilopoiesis.

Eosinophils were originally thought to be end-stage cells mainly involved in host protection from parasites. In addition to this traditional role, more diverse eosinophil functions have been identified including antibacterial and anti-fungal activities[1,2], regulation of other immune cell populations[3], maintenance of tissue homeostasis[4,5], and regeneration of liver and muscle[6,7]. Eosinophils also contribute to the pathogenesis of eosinophil-associated respiratory and gastrointestinal disorders[8]. Hence, analysis of the pattern and regulation of eosinophilopoiesis in physiologic and pathologic conditions is essential to understanding the control of eosinophil populations and their distribution.

[1]Department of Hematology, the Second Affiliated Hospital, School of Medicine, South China University of Technology, Guangzhou 510006, P.R. China. [2]Innovation Centre of Ministry of Education for Development and Diseases, School of Medicine, South China University of Technology, Guangzhou 510006, P.R. China. [3]Singapore Immunology Network (SIgN), A*STAR (Agency for Science, Technology and Research), Biopolis 138648, Singapore. [4]Department of Developmental Biology, School of Basic Medical Sciences, Southern Medical University, Guangzhou 510515, P.R. China. [5]Shanghai Institute of Immunology, Shanghai JiaoTong University School of Medicine, Shanghai, P.R. China. [6]Sun Yat-Sen University Cancer Center, State Key Laboratory of Oncology in South China, Collaborative Innovation Center for Cancer Medicine, Guangzhou, P.R. China. [7]Shanghai Immune Therapy Institute, Shanghai Jiao Tong University School of Medicine Affiliated Renji Hospital, Shanghai, P.R. China. [8]Department of Microbiology and Immunology, Immunology Translational Research Program, Yong Loo Lin School of Medicine, Immunology Program, Life Sciences Institute, National University of Singapore, Singapore 117543, Singapore. [9]These authors contributed equally: Gaofei Li, Yicong Sun. ✉e-mail: nglaiguan@renji.com; mczhangyy@scut.edu.cn

During adult hematopoiesis, eosinophils are known to be derived from eosinophil lineage-committed progenitors (EoP), which are generated in the bone marrow (BM) from common myeloid progenitors in humans and granulocyte/macrophage progenitors in mice[9,10]. However, the spatial-temporal development of eosinophil lineage cells is largely unknown as the study of eosinophil development during early life in mammalian models is limited.

The regulation of eosinophil lineage development is tightly controlled by extrinsic and intrinsic factors. Several cytokines (IL-5, IL-3, and GM-CSF) promote eosinophil development[11]. Interestingly, mutations of these cytokines do not affect eosinophil levels[12–14], suggesting that these extrinsic factors are not essential steady-state regulators. Transcription factors (TFs) play critical roles in eosinophilopoiesis, with the GATA family of TFs essential to this process[15,16]. Several TFs (CCAAT enhancer-binding protein (C/EBP) and Ets family members) are involved in myelopoiesis and are thought to be involved in eosinophilopoiesis[17–19], though their precise roles in eosinophilopoiesis are not clearly defined. It is, therefore, essential to analyze the step-wise transcriptional regulation of eosinophil development as a means by which to better understand eosinophilopoiesis.

Zebrafish is an ideal model for the exploration of both normal and malignant hematopoiesis[20], as blood cell development and associated cellular functions are very similar to that of mammals. Zebrafish eosinophils respond to parasites in a manner similar to that of mammalian eosinophils[21] as well as to some bacteria[22–24]. Adult zebrafish eosinophils were found not only in the kidney but also in the peritoneal fluid and intestine[21], while their early distribution remains unclear. The *Tg(gata2a:eGFP)* zebrafish line labeled adult zebrafish eosinophils[21], but did not identify eosinophils during the early stages of development[25,26]. Therefore, it is important to develop a specific eosinophil marker from the earliest developmental stage through adulthood on the whole organism level.

In this study, we depict the spatial-temporal development of eosinophils during early development by utilizing an eosinophil-specific marker, *si:dkeyp-75b4.10* (*eslec*), which specifically labels zebrafish eosinophils from early life through adulthood. By identification of the eosinophil transcriptional profile, the landscape of zebrafish eosinophil lineage is clarified. Moreover, the roles of C/EBP family TFs in shaping eosinophil lineage development and species convergence are demonstrated. Taken together, we decipher the developmental pattern and regulation of eosinophils, providing new insights into the biology of eosinophils.

## Results

### *Eslec* specifically labels zebrafish eosinophilic cells

Eosinophils are characterized in adult zebrafish[21], but their developmental origins remain unclear due to a lack of lineage-restricted markers. To identify genes that specifically mark the eosinophil lineage from early life to adulthood, we reanalyzed a single-cell RNA-Seq (scRNA-Seq) dataset containing zebrafish kidney cells[27]. The zebrafish kidney marrow (KM) is analogous to the mammalian BM[20]. The top marker gene expressed in the *gata2a*+ cluster (eosinophils[21]) was *si:dkeyp-75b4.10* (Fig. S1A, B), exhibiting a specific and robust expression in eosinophils (Fig. S1C). This gene is predicted to encode a C-type lectin (Fig. S1D), belonging to the same protein family as the mammalian eosinophil major basic protein[28]. Therefore, *si:dkeyp-75b4.10* (renamed as *eslec*, eosinophil-specific lectin, for short) was investigated as a potential eosinophil marker.

To assess the dynamic development of *eslec*+ cells, we then established two transgenic reporter lines, *Tg(eslec:eGFP)* and *Tg(eslec:dsRed)*, with the fluorescent proteins driven by the 4.3 kb *eslec* promoter (Fig. 1A). Consistent with the detection of *eslec*+ cells in whole-mount in-situ hybridization (WISH) (Fig. S1E), *eslec*:eGFP+ cells were undetectable at 4 days post fertilization (dpf), but emerged from 5 dpf onwards, with increased detection at 7 dpf (Fig. 1B–C). The

fluorescence-activated cell sorting (FACS)-sorted *eslec*:eGFP+ cells from larvae exhibited typical zebrafish eosinophil morphologies (Fig. 1D), with the nucleus eccentrically located as previously reported[29]. Unlike *gata2a*:eGFP+ cells broadly marking neural cells and vessels (Fig. S1F), the *eslec*:DsRed+ cells were spottily distributed in the visceral region and only colocalized with a small population of *gata2a*:eGFP+ cells (Fig. S1F), indicating that *eslec* is more specific for eosinophils than *gata2a*. Moreover, the *eslec*:eGFP+ cells belonged to *coro1a*+ cells (which marks all myeloid cells[30]), but were not colocalized with *lyz*+ neutrophils[31] or *mpeg1*+ macrophages[32] (Fig. S1G), confirming the lineage specificity of *eslec*. With *cdh17*:DsRed+ fluorescent indication for the visceral region[33], *eslec*:eGFP+ eosinophils were found in the kidney, intestine, and a probably intraperitoneal location as early as 5 dpf and abundantly within 7 dpf larvae (Fig. 1E), suggesting that eosinophilic cells are distributed in these tissues during early life.

Next, we explored whether the *Tg(eslec:eGFP)* transgenic line could label eosinophils throughout the lifespan of zebrafish. Using flow cytometry analysis, we detected the percentages of *eslec*+ eosinophils at different developmental stages (Fig. S2A–B). Similar to the above results (Fig. 1B–C), eosinophils were rarely detected in 4 dpf larvae, but emerged after 5 dpf and increased significantly in 7 dpf larvae (Fig. S2A–B). During the late-larval stage (21 dpf) and juvenile stage (49 dpf), eosinophils could be detected but with lower percentages compared with that of 7 dpf larvae (Fig. S2A–B), which might be due to the faster somatic growth than eosinophilopoiesis during these stages. In adult zebrafish, eosinophils could be found abundantly (>1%) in the KM and intraperitoneal exudate (IPEX; Fig. S2C–D), as reported before[21]. Besides, we also found eosinophils in the intestine (~0.5%), spleen (~0.3%), and liver (~0.1%), while eosinophils were few in the peripheral blood, brain, and heart (<0.01%; Fig. S2C–D). Together, we conclude that *eslec* is a bona fide eosinophil marker that can be used to identify zebrafish eosinophils from early life to adulthood, demonstrating the spatial-temporal development pattern of zebrafish *eslec*+ eosinophils.

### ScRNA-Seq defines the eosinophil lineage

To further decipher the developmental and functional features of the eosinophil lineage, we performed scRNA-Seq analysis with the *eslec*:eGFP+ cells of adult zebrafish kidneys, whose morphologies are similar with *gata2a*:eGFP+ KM eosinophils[29] and larval eosinophils (Fig. 2A). The high expression of *eslec* could be detected in each cell, in which *coro1a* was detectable but *lyz* and *mpeg1* were not expressed (Fig. 2B and Fig. S3A), indicating that all analyzed cells belong to the eosinophil lineage. As both S-phase and G2/M-phase proliferating cells could be found through the in silico cell cycle analysis, a partial cell cycle regression was performed to regress their differences (Fig. 2C). After the regression, the KM eosinophil lineage cells were further clustered into three major clusters, namely Cluster1, Cluster2, and Cluster3 (Fig. 2D).

Based on Gene Ontology (GO)/Kyoto Encyclopedia of Genes and Genomes (KEGG) annotations and the expression of several key genes (Supplementary Data 1), we further assigned these clusters to eosinophils at different stages. Firstly, Cluster1 showed a strong proliferating trend (Fig. S3B), with genes such as *mki67*, *top2a*, *pcna*, and *mcm2* highly expressed (Fig. S3C). Therefore, Cluster1 was renamed as EoPs, as the hematopoietic progenitors are predominantly more proliferative than mature leukocytes[34]. On the other hand, the mature eosinophils (mat.Eos) were assigned to Cluster3 based on the expression of multiple immune responses (Fig. S3D) and several immune-related receptors such as *cxcr4b*, *ccr9a*, *fcer1gl*, and *il1rl1* (Fig. S3E). Finally, the intermediate cluster (Cluster2) between EoPs and mat.Eos was therefore regarded as eosinophil precursors (pre.Eos). The pre.Eos exhibited high expression of genes related to ribosome, oxidative phosphorylation (*cox5ab* and *cox6a2*) and protein processing in endoplasmic reticulum (ER) (*agr2* and *pdia2*) (Fig. S3F–G), which is in line with the activation of ER stress pathway during eosinophil maturation[35].

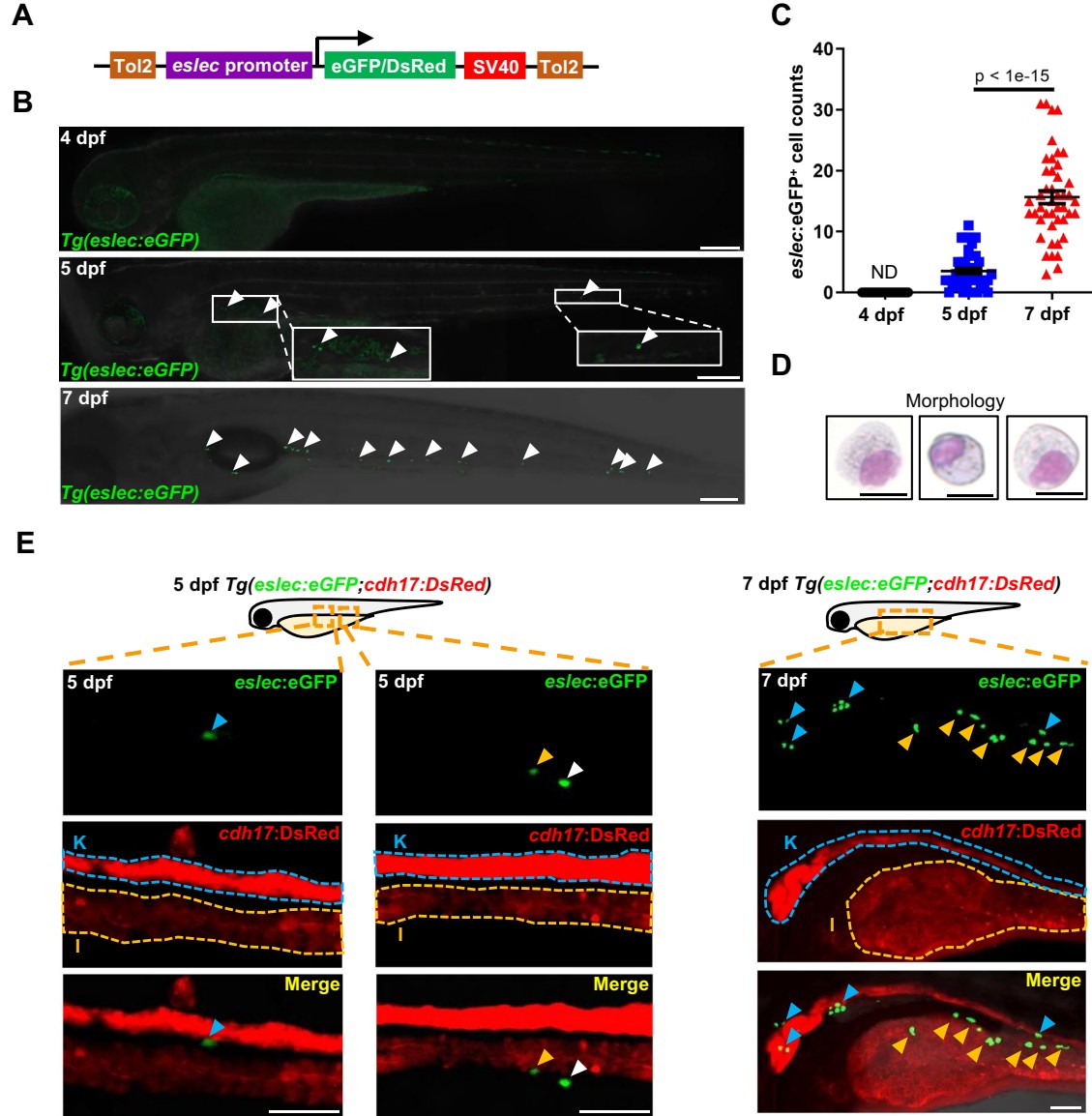

**Fig. 1 | The early spatial-temporal development and function of zebrafish eosinophils. A** Design of vector for generating *Tg(eslec:eGFP)* or *Tg(eslec:DsRed)* line. The eGFP or DsRed fluorescence is driven by the 4.3 kb *eslec* promoter and terminated by the SV40 terminator. Tol2 system was used to integrate these elements into the zebrafish genome. **B** Temporal development pattern of early eosinophils. No eosinophil could be found in 4 dpf larvae, while they start to emerge from 5 dpf onwards and increase at 7 dpf (white arrowhead, bar = 200 μm, white box showing enlarged view). **C** Quantification of **B**. The eosinophil populations were analyzed by Student's *t*-test (two-sided, mean ± SEM, ND means not detectable).

Sample size: 4 dpf, *n* = 30; 5 dpf, *n* = 45; 7 dpf, *n* = 45. Three independent experiments were conducted. **D** Morphologies of *eslec*⁺ cells. The eGFP⁺ cells were sorted from 7 dpf *Tg(eslec:eGFP)* larvae and applied to May-Giemsa staining (bar = 10 μm). Two independent experiments were conducted. **E** Spatial development pattern of early eosinophils. The kidney and intestine are respectively marked by "K" or "I" and surrounded by dotted lines, based on the *cdh17* expression. The eosinophils located in the kidney, intestine, or possibly peritoneal fluid are respectively marked by blue, yellow, or white arrowhead (bar = 50 μm). Two independent experiments were conducted. Source data are provided as a Source Data file.

Therefore, it could be concluded that the *eslec*⁺ cells consist of an eosinophil lineage from the EoPs to mature eosinophils. With Monocle3 and RNA velocity, we analyzed the trajectory of the eosinophil lineage, which delineated a presumed developmental orientation from EoP towards pre.Eos and mat.Eos thereafter (Fig. 2E, F). Taken together, we delineate the eosinophil lineage cells into three developmental stages: EoPs, pre.Eos, and mat.Eos (Fig. 2G).

**Cebp1 inhibits eosinophil lineage commitment**

The specification and commitment of eosinophil lineage have been largely attributed to the TF C/EBPε in mice[17]. We next wondered if zebrafish eosinophilopoiesis could be regulated in a similar manner.

We previously found that the mutation of zebrafish *cebp1*, the orthologue of mammalian *Cebpe*, severely impaired neutrophil differentiation without affecting myeloid progenitors and macrophages[36]. In eosinophils, we found that *eslec*⁺ cells were significantly increased in the *cebp1*^smu1^ (*cebp1*⁻/⁻ hereafter) mutants through WISH analysis (Fig. 3A). Furthermore, over-expression of *cebp1* severely inhibited eosinophil development (Fig. 3B). In adult fish, the eosinophil percentages were also higher in the KM, IPEX, intestine, spleen, and liver in *cebp1* mutants (Fig. S4). These results signify that *cebp1* is a critical regulator of eosinophilopoiesis.

To further determine the effect of *cebp1* on hematopoiesis, we performed a scRNA-Seq analysis of whole KM cells of *cebp1* mutants and WT zebrafish (Fig. 3C and S5A). The integrated uniform manifold

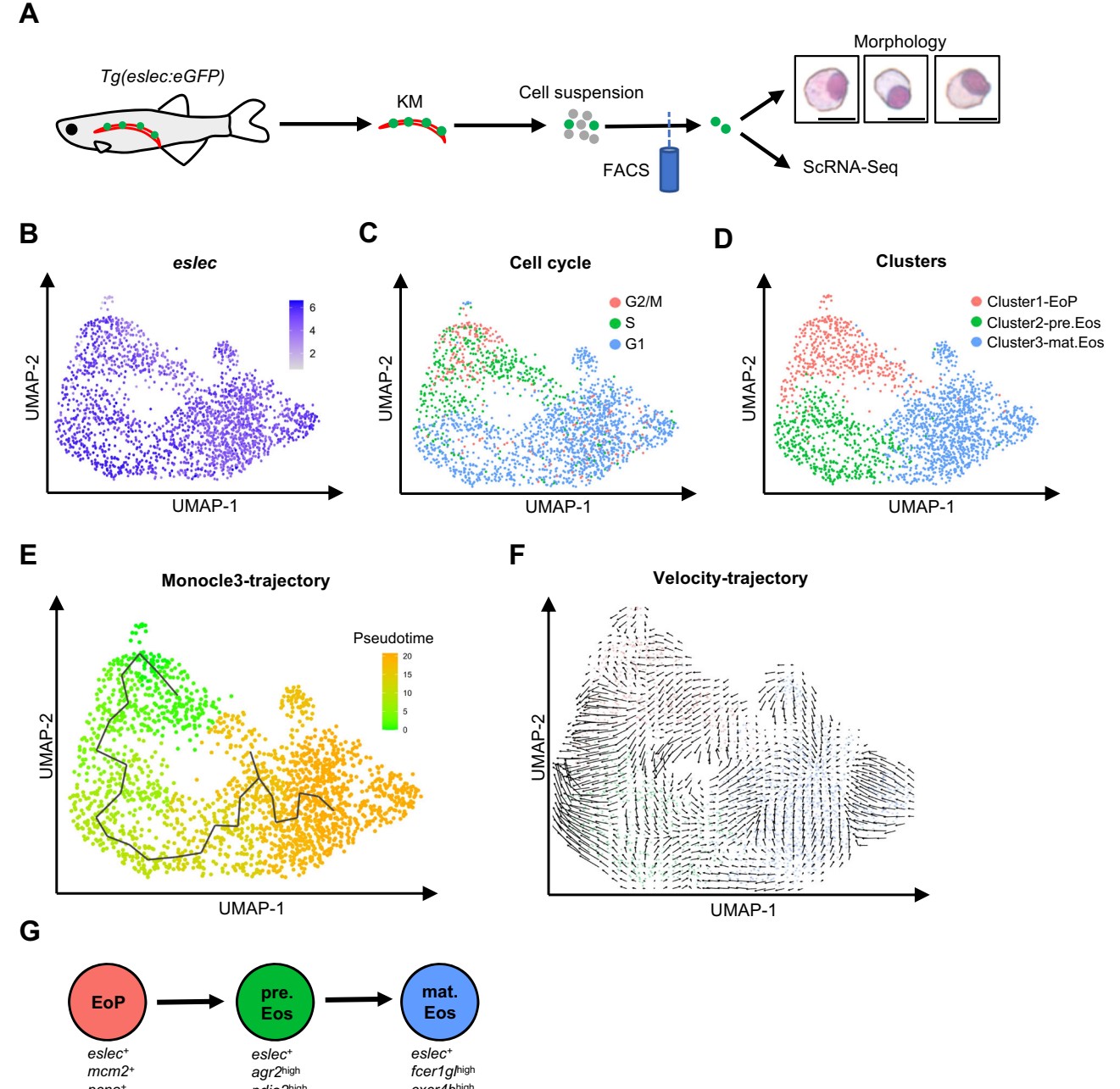

**Fig. 2 | Landscape of the eosinophilic cell lineage. A** Design of the scRNA-Seq analysis. Adult *Tg(eslec:eGFP)* kidneys were dissected and pipette-mixed into cell suspensions. The eGFP⁺ cells were sorted and stained with May-Giemsa to check their morphologies (bar = 10 μm, two independent experiments were conducted). Finally, the *eslec*:eGFP⁺ cells were applied to scRNA-Seq analysis. **B** UMAP showing that all analyzed cells were *eslec*⁺. **C** UMAP showing the cell cycle status of KM eosinophilic cells. **D** UMAP showing the clustering information of KM eosinophilic cell. **E** Monocle3-trajectory of the eosinophil lineage. Pseudotime status of eosinophils were calculated with Monocle3. **F** Velocity-trajectory of the eosinophil lineage. RNA velocity stream plot shows the directional flow of eosinophils. **G** Model of the eosinophilic cell lineage. The KM eosinophils are sub-clustered into three stages, namely EoP (*eslec*⁺*mcm2*⁺*pcna*⁺), pre.Eos (*eslec*⁺*agr2*high*pdia2*high), and mat.Eos (*eslec*⁺*fcer1gl*high*cxcr4b*high).

approximation and projection (UMAP) demonstrated that the neutrophil lineage was almost absent in the KM of the *cebp1* mutant (Fig. 3C), confirming the requirement of Cebp1 for neutrophil development. Instead, we found an atypical cell population in *cebp1*⁻/⁻ KM with proliferation features (Fig. 3C and S5B). For eosinophils, we found the percentage increased in the *cebp1* mutant (2.94 ± 0.15%) compared to WT (1.19 ± 0.26%; Figs. 3C and S5C). We further normalized the eosinophil populations with hematopoietic stem/progenitor cells (HSPC) (unaltered upon mutation, Fig. S5D) and found the relative eosinophil percentage still significantly higher in the *cebp1* mutant (Fig. S5E).

Based on the KM *eslec*:eGFP⁺ cell information (Fig. 2), we further divided the integrated eosinophil lineage into subclusters: EoP (*mcm2*⁺ and *pcna*⁺), pre.Eos (*agr2*high and *pdia2*high), and mat.Eos (*fcer1gl*high and *cxcr4b*high; Fig. 3C–D). In addition, we identified a cluster of *eslec*low cells with high expression of HSPC markers (*ncf2* and *myca*), linking the *eslec*⁺ population with the HSPCs in the UMAP (Fig. 3C–D). Therefore, we defined this cluster as eosinophil lineage-biased progenitors (Eb-P). In the *cebp1* mutant, we found that the Eb-P percentages were significantly higher compared to WT, while the EoP also exhibited an increasing trend (Fig. 3E), indicating that *cebp1* acts as a negative regulator in eosinophil fate commitment. Contrastingly, *cebp1* mutants had slightly

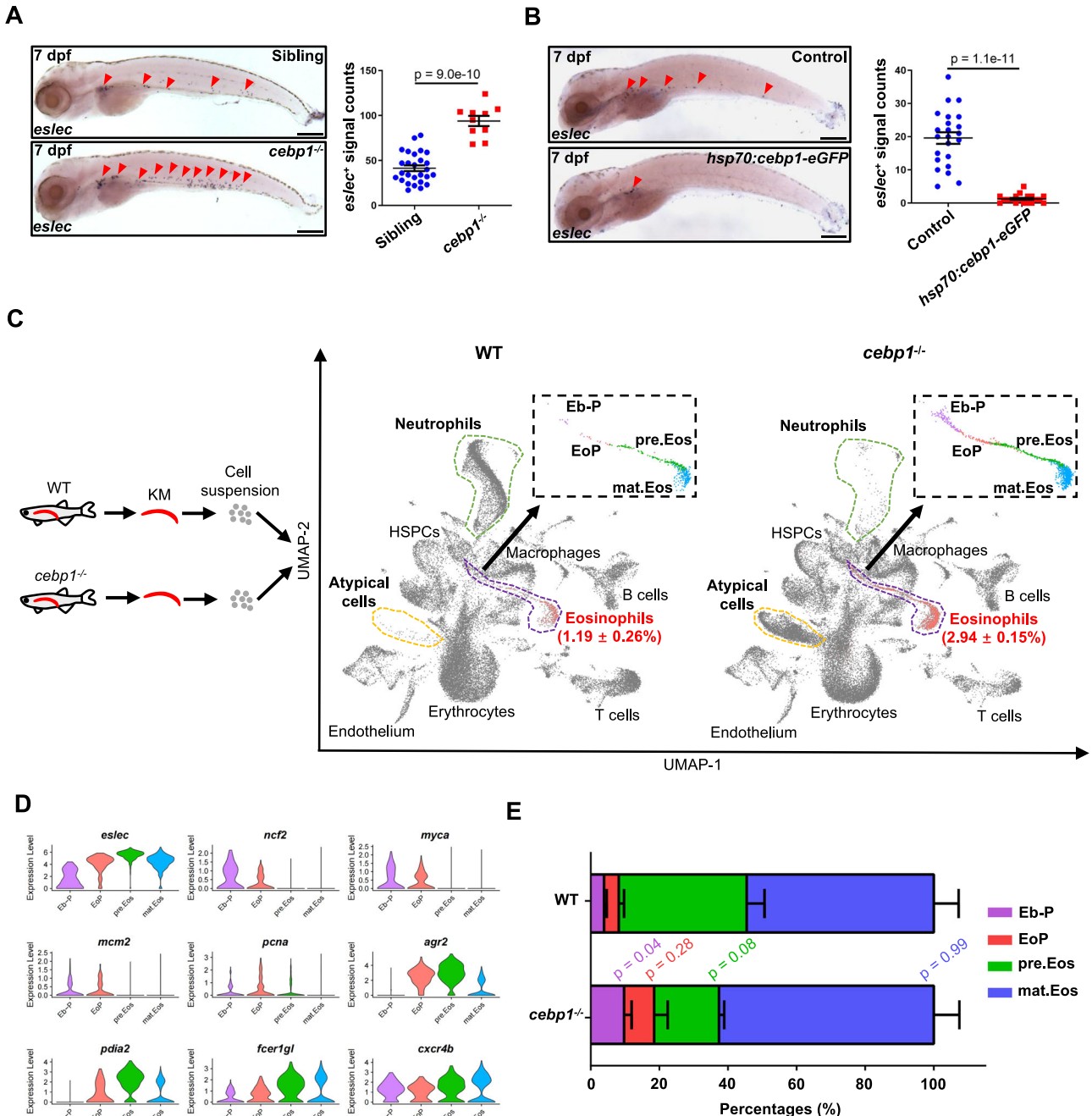

**Fig. 3 | Cebp1 inhibits the commitment of eosinophils. A** Mutation of *cebp1* enhanced eosinophilopoiesis. Eosinophil (red arrowhead) numbers were increased in 7 dpf *cebp1* mutant larvae (bar = 200 μm). Quantification was performed using Student's *t*-test (two-sided, mean ± SEM). Sample size: Sibling, *n* = 27; *cebp1⁻/⁻*, *n* = 10. Three independent experiments were conducted. **B** Overexpression of *cebp1* inhibited eosinophilopoiesis. Eosinophil (red arrowhead) numbers were decreased in 7 dpf *Tg(hsp70:cebp1-eGFP)* larvae (bar = 200 μm). Quantification was performed using Student's *t* test (two-sided, mean ± SEM). Sample size: Control, *n* = 23; *hsp70:cebp1-eGFP*, *n* = 19. Three independent experiments were conducted. **C** UMAP showing the scRNA-Seq clustering information. KM cells of WT and *cebp1⁻/⁻* zebrafish were integrated and analyzed by scRNA-Seq analysis. The purple, green,

and yellow dotted line-surrounding regions respectively indicate the eosinophil lineage, neutrophil lineage, and atypical cell cluster of each sample. Eosinophil lineage cells of each sample were further subclustered and the enlarged view (black-boxed region) is shown. **D** Subcluster annotation reference. The expression patterns of *eslec* and the markers of each cluster (Eb-P: *ncf2* and *myca*; EoP: *mcm2* and *pcna*; pre.Eos: *agr2* and *pdia2*; mat.Eos: *fcer1gl* and *cxcr4b*) are shown by violin plots. **E** Eosinophil lineage profile with *cebp1* mutation. Percentages of eosinophil lineage cells of different stages were analyzed in WT and *cebp1⁻/⁻* zebrafish (*n* = 3, Student's *t* test, two-sided, mean ± SEM). Source data are provided as a Source Data file.

lower percentages of pre.Eos (Fig. 3E), suggesting that Cebp1 may potentially influence eosinophil differentiation. In summary, our data demonstrate that zebrafish Cebp1 is indispensable for neutrophil lineage development and the suppression of eosinophil lineage commitment, with possible participation in eosinophilic cell differentiation.

## Cebp1 represses *cebpb* to inhibit eosinophil lineage commitment

As Cebp1 negatively regulates eosinophilopoiesis, we hypothesized that Cebp1 might transcriptionally suppress a series of eosinophil differentiation/maturation-related functional genes, or directly inhibit

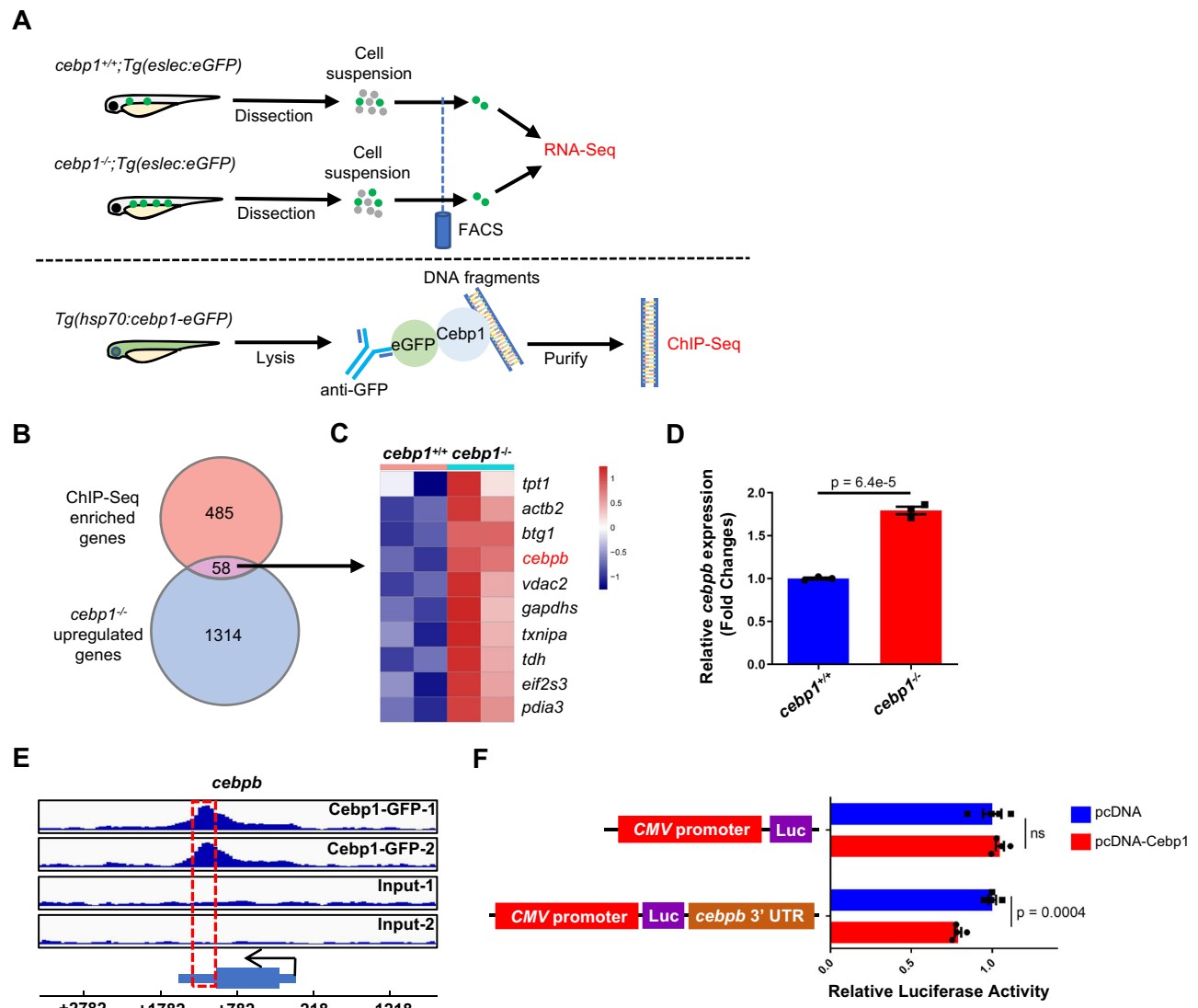

**Fig. 4 | *Cebpb* is directly suppressed by Cebp1. A** Design of Cebp1-target screening. RNA-Seq was performed with *cebp1*⁺/⁺ and *cebp1*⁻/⁻ eosinophils to identify upregulated genes with *cebp1* mutation (1,372 genes, q < 0.05, baseMean > 50, log2 fold change > 0.7). ChIP-Seq was performed with *Tg(hsp70:cebp1-eGFP)* larvae to identify Cebp1-binding genes (543 genes, q < 0.05, peak fold enrich > 2, TSS ± 2 kb). **B** GO/KEGG analysis of overlapping genes. Fifty-eight genes were found to overlap in the Cebp1-binding genes and upregulated genes of *cebp1*⁻/⁻ eosinophils. **C** Heatmap showing the top 10 highest expressed genes of all the overlapping genes. **D** RT-qPCR showing the relative *cebpb* expression in WT and *cebp1*⁻/⁻ eosi- nophils. (Student's *t* test, two-sided, mean ± SEM). Three independent experiments

were conducted with *n* = 3 in each group. **E** Cebp1 binding on the 3′ UTR of *cebpb*. At the bottom panel, the direction of *cebpb* transcription is marked with a black arrow, and the broad box and narrow box represent the coding sequence and UTR region, respectively. The red dotted box shows the binding peaks. **F** Cebp1 inhibiting the expression of *cebpb*. Luciferase assay demonstrated that Cebp1 inhibited the luci- ferase activities when the 3′UTR of *cebpb* was added after the luciferase gene (Luc) (Student's *t* test, two-sided, mean ± SEM, ns represents no significance). Three independent experiments were conducted with *n* = 4 in each group. Source data are provided as a Source Data file.

eosinophil lineage TFs. To test our hypothesis, we combined both bulk RNA-Seq and chromatin immune precipitation (ChIP)-Seq to identify the downstream target genes of Cebp1, which are upregulated by *cebp1* mutation through the direct binding of Cebp1. Bulk RNA-Seq was performed with sorted eosinophils from WT or *cebp1* mutant larvae (Fig. 4A). A sample-to-sample distance heatmap indicated that the differences between groups were much more significant than those within groups (Fig. S6A). ChIP-Seq was performed with anti-GFP antibody and lysates of 5 dpf *Tg(hsp70:cebp1-eGFP)* larvae in which heat-shock induced Cebp1-eGFP fusion proteins were expressed (Fig. 4A). Cebp1-binding DNA sequence analysis found 14% interaction in promoter regions and 5.8% in exons and untranslated regions (UTRs) (Fig. S6B). By the combination of both datasets, 58 genes were found to be highly expressed in *cebp1*⁻/⁻ eosinophils (bulk RNA-Seq)

and also with binding of Cebp1 (ChIP-Seq; Fig. 4B), suggesting that these genes may be directly inhibited by Cebp1.

We sorted all the overlapping genes by expression level (Supple- mentary Data 2) and assessed the top 10 genes by RNA-Seq heatmap (Fig. 4C). Among these genes, the myeloid TF *cebpb* has the strongest ChIP-binding peak intensity of all overlapping genes (Supplementary Data 3). Through RT-qPCR assays, we validated that *cebpb* was upre- gulated in eosinophils upon *cebp1* mutation (Fig. 4D). The CEBP family TF, Cebpβ, is known to be involved in emergency myelopoiesis rather than normal neutrophil development[37,38], while its in vivo function in eosinophilopoiesis remains unclear. The *cebpb* gene interaction region for Cebp1 was at the 3′UTR (Fig. 4E), further suggesting that Cebp1 likely inhibits the expression of *cebpb*. To verify that *cebpb* is tran- scriptionally regulated by Cebp1, we performed a dual-luciferase assay

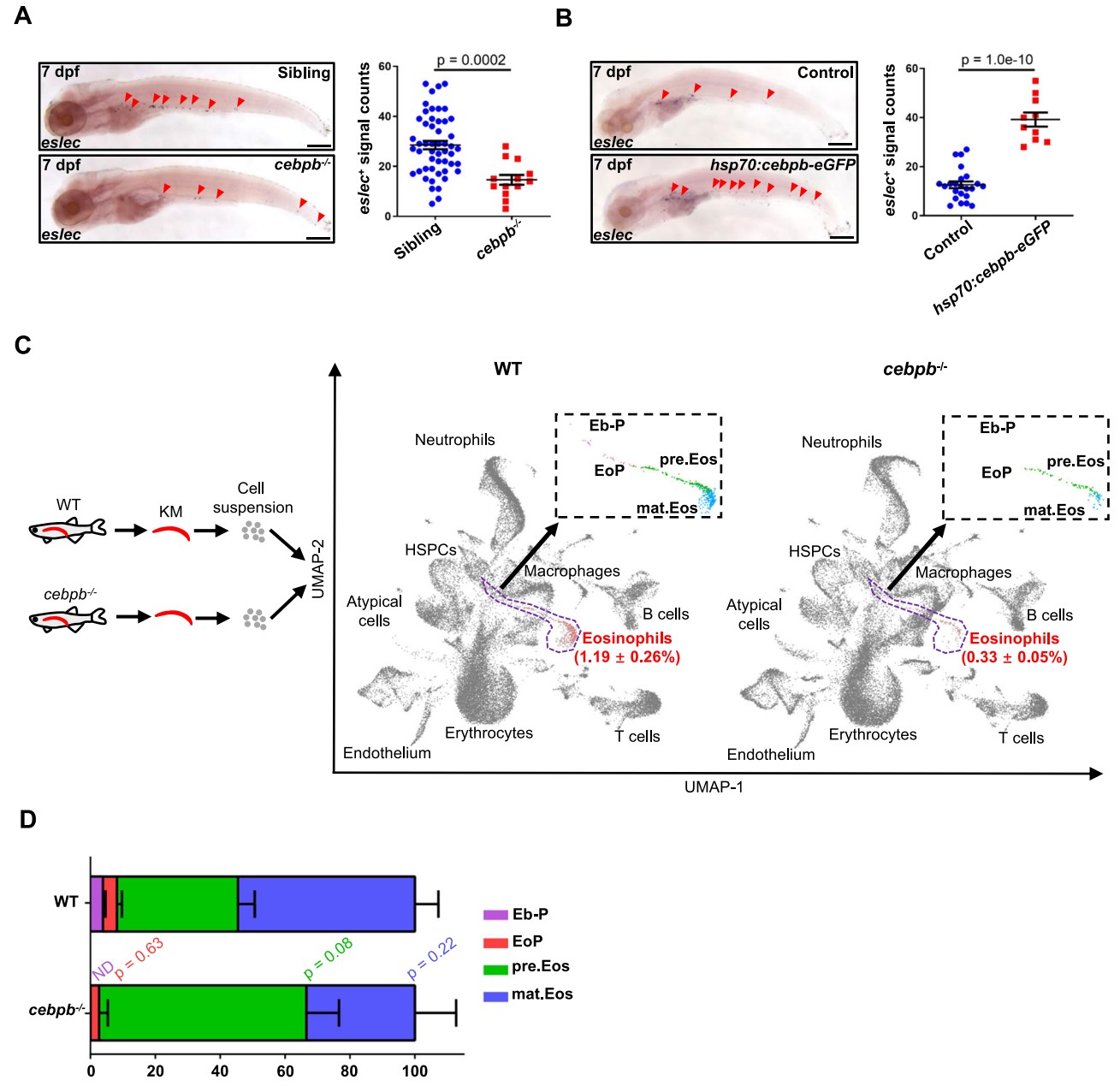

**Fig. 5 | Cebpβ promotes the commitment of the eosinophil lineage. A** Cebpβ is a positive regulator of eosinophilopoiesis. Eosinophil (red arrowhead) numbers were found to decrease in 7 dpf *cebpb* mutant larvae (bar = 200 μm). Quantification was performed using Student's *t*-test (two-sided, mean ± SEM). Sample size: Sibling, *n* = 51; *cebpb⁻/⁻*, *n* = 13. Three independent experiments were conducted.
**B** Overexpression of *cebpb* enhanced eosinophilopoiesis. Eosinophil (red arrowhead) numbers were increased in 7 dpf *Tg(hsp70:cebpb-eGFP)* larvae (bar = 200 μm). Quantification was performed using Student's *t* test (two-sided, mean ± SEM). Sample size: Control, *n* = 23; *hsp70:cebpb-eGFP*, *n* = 10. Three independent

experiments were conducted. **C** UMAP showing the scRNA-Seq clustering information. KM cells of WT and *cebpb⁻/⁻* zebrafish were integrated and analyzed by scRNA-Seq analysis. The dotted line-surrounding regions indicate the eosinophil lineage of each sample, which were further subclustered and the enlarged view is shown. **D** Eosinophil lineage profile with *cebpb* mutation. Percentages of eosinophil lineage cells of different stages were analyzed in the KM of WT and *cebpb⁻/⁻* zebrafish (*n* = 3, Student's *t* test, two-sided, mean ± SEM, ND means not detectable). Source data are provided as a Source Data file.

in 293 T cells. Overexpression of Cebp1 repressed the luciferase activity of cells transfected with the pMIR-Report-*cebpb*-3′ UTR vector (Fig. 4F). These results demonstrate that Cebp1 directly interacts with the 3′UTR region of *cebpb*, inhibiting the expression of *cebpb*.

We examined our previously generated *cebpb^szy7* (*cebpb⁻/⁻* hereafter) mutant[38] and found eosinophil populations to be significantly decreased in *cebpb⁻/⁻* larvae (Fig. 5A), which mimics the eosinopenia in Cebp1 overexpression. We also generated a transgenic line, *Tg(hsp70:cebpb-eGFP)*, for the heat-shock-induced overexpression of

Cebpβ-eGFP fusion protein. We found that overexpression of Cebpβ significantly enhanced eosinophilopoiesis (Fig. 5B), confirming the role of Cebpβ in eosinophilopoiesis promotion. In adult fish, the eosinophil percentages were also significantly lower in the KM, IPEX, intestine, spleen, and liver of *cebpb* mutants (Fig. S7). These results confirmed Cebpβ as a negative regulator of eosinophilopoiesis,

To further delineate the effects of *cebpb* on eosinophilopoiesis, we performed scRNA-Seq analysis with whole KM cells of the *cebpb* mutants[38]. Integrated UMAP analysis showed a significant decrease in

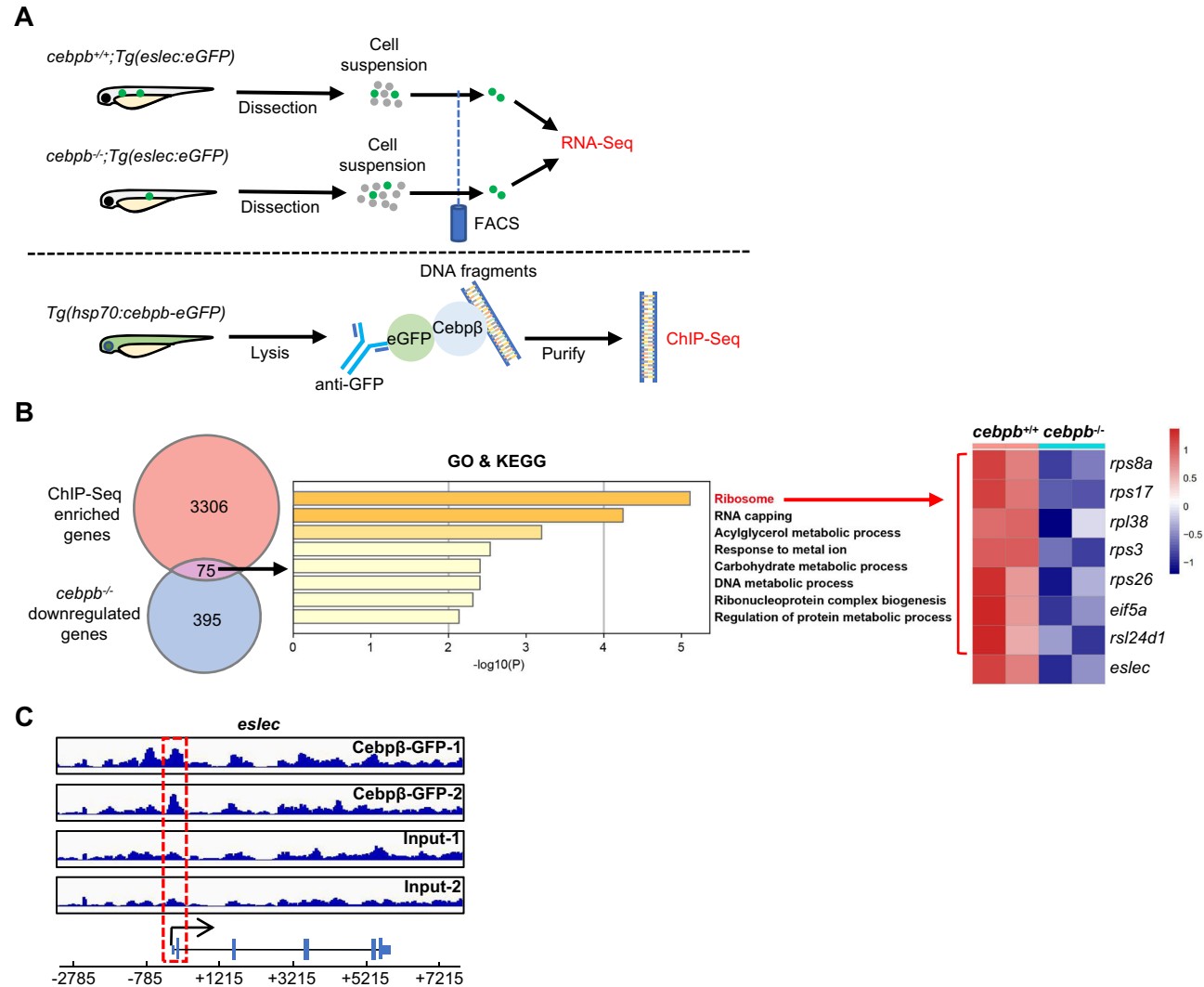

**Fig. 6 | Cebpβ promotes eosinophilopoiesis by regulating pre.Eos-related genes. A** Design of Cebpβ-target screening. RNA-Seq was performed with *cebpb⁺/⁺* and *cebpb⁻/⁻* eosinophils to identify downregulated genes with *cebpb* mutation (470 genes, *q* < 0.05, baseMean > 50, log2 fold change > 0.7). ChIP-Seq was performed with *Tg(hsp70:cebp1-eGFP)* larvae to identify Cebpβ-binding genes (3381 genes, *q* < 0.05, peak fold enrich > 2, TSS ± 2 kb). **B** Downstream targets of Cebpβ. Seventy-five genes were found overlapped from the Cebpβ-binding genes and the downregulated genes with *cebpb* mutation. GO/KEGG analysis suggested that these genes are associated with ribosome (Red). The genes related to this GO term and *eslec* were exhibited by RNA-Seq heatmap. **C** Cebpβ binding on the proximal promoter of *eslec*. At the bottom panel, the direction of *eslec* transcription is marked with a black arrow, and the broad box and narrow box represent the coding sequence and UTR region, respectively. The red dotted box shows the binding peaks.

the eosinophil population in the *cebpb* mutant (0.33 ± 0.05%) compared to WT (Figs. 5C and S4C–E). The *cebpb* mutant zebrafish had no detectable Eb-P, while showed an increased tendency of pre.Eos and a decreased tendency of mat.Eos compared to WT subclusters (Fig. 5C–D). Therefore, our results demonstrated the impairment of eosinophil lineage commitment and suggested the developmental arrest at pre.Eos stage by *cebpb* deficiency.

### Cebpβ induces eosinophilopoiesis through pre.Eos-related genes

As Cebpβ promotes eosinophilopoiesis, we hypothesized that Cebpβ may transcriptionally facilitate downstream differentiation-related genes and factors required for lineage development by direct binding of Cebpβ, which genes are consequentially downregulated by *cebpb* mutation. To test this, we analyzed the downstream targets of Cebpβ on eosinophilopoiesis by bulk RNA-Seq (Fig. S6C and Supplementary Data 4) of sorted eosinophil lineage cells from WT and *cebpb⁻/⁻* larvae and ChIP-Seq (Fig. S6D and Supplementary Data 5) assessment of

*Tg(hsp70:cebpb-eGFP)* larvae (Fig. 6A). Seventy-five genes were found to overlap in downregulated genes of *cebpb* mutant eosinophilic cells and Cebpβ-binding genes, suggesting that these genes may be directly activated by Cebpβ (Fig. 6B). GO/KEGG analysis of these genes showed that ribosome-related genes were enriched (Fig. 6B), including *rps8a*, *rps17*, *rpl38*, *rps3*, *rps26*, *eif5a*, and *rsl24d1*. In addition, *eslec* was also found to be elevated by Cebpβ, which could bind to the proximal promoter of *eslec* (Fig. 6B–C). As *eslec* and ribosome-related genes were found highly expressed in pre.Eos (Figs. 3D and S3G), it suggests that Cebpβ directly regulates a series of downstream genes related to pre.Eos. Combined with the scRNA-Seq data, Cebpβ probably exerts its function in eosinophil lineage cell differentiation by regulating pre.Eos-related genes.

### Conserved and divergent roles of C/EBP across species

To further assess the evolutionary conservation and divergence of CEBPs in regulating granulocyte development, we evaluated and compared our findings with those from *Cebpe* knockout mice. As

previously reported, complete disruption of mouse *Cebpe* resulted in the absence of both neutrophils and eosinophils, an unaffected monocytic lineage, and the appearance of an atypical Siglec-F⁺CD115⁺ population[39] (reanalyzed and shown in Fig. 7A). It led to a question whether these atypical cells are associated with eosinophils, based on their Siglec-F⁺ feature. Using a neutrophil-specific cre-mediated *Cebpe* conditional knockout mouse model (*Mrp8*ᶜʳᵉ*Cebpe*ᶠˡ/ᶠˡ), we found that eosinophil development was normal but neutrophil development was blocked at the pre.Neu stage (Fig. 7B). Consistent with the *Cebpe*-deficient mice, we observed a similar appearance of atypical granulocytes (Fig. 7B), suggesting that they are derived at the expense of neutrophils but not eosinophils. Thus, although mouse C/EBPε and Cebp1 have convergent roles in regulating neutrophil development, their effects on eosinophilopoiesis are divergent.

In humans, there are four isoforms of the CEBPε protein, two of which (C/EBPεᵖ²⁷ and C/EBPεᵖ¹⁴) play repressive roles in eosinophilopoiesis, while C/EBPεᵖ³² and C/EBPεᵖ³⁰ induce eosinophil development[40–42]. Therefore, we detected whether human C/EBPε isoforms could inhibit zebrafish eosinophilopoiesis, by utilizing the *coro1a* promoter to induce myeloid-specific C/EBPε-eGFP fusion gene overexpression. The fusion proteins were validated not to affect the TF activity of C/EBPs, by a luciferase assay showing their transactivation activity on *cebpb* or *MBP-P2*, which is reported as a direct target of C/EBPε[41] (Fig. S8). As a control, overexpression of zebrafish Cebp1-eGFP strongly inhibited zebrafish eosinophilopoiesis (Fig. 7C), consistent with the ubiquitous overexpression data (Fig. 3B). We found that among all the C/EBPε isoforms, only human C/EBPεᵖ²⁷ significantly repressed zebrafish eosinophilopoiesis, while others did not affect eosinophil development (Fig. 7C). Therefore, our data demonstrate that Cebp1 is the functional orthologue of human C/EBPεᵖ²⁷ in repressing eosinophil development.

## Discussion

In this study, we identify *eslec* as a reliable zebrafish eosinophil lineage-specific marker, and generate a *Tg(eslec:eGFP)* zebrafish line that allows us to understand the development of eosinophils from early life to adulthood. In this manner, the spatial-temporal developmental pattern, lineage landscape, and genetic regulation of eosinophils are determined. We demonstrate the importance of Cebp1 in balancing granulopoiesis and clarify the interplay of Cebp1 and Cebpβ during eosinophilopoiesis. In zebrafish, Cebp1 is not only required for neutrophil development, but also regulates the balance between neutrophil and eosinophil commitment. Furthermore, the direct regulation of Cebp1 on Cebpβ determines eosinophil lineage differentiation (Fig. 8). We further demonstrate that zebrafish Cebp1 has divergent effects with mouse C/EBPε, but convergent roles with human C/EBPεᵖ²⁷ in eosinophilopoiesis. Reported results provide a new understanding of eosinophil lineage development.

The eosinophil marker *eslec* was identified by a combination of scRNA-Seq and experimental validation. Recently, a mycobacterial infection scRNA-Seq dataset employed *itln1* as a zebrafish eosinophil marker[24]. By reanalyzing the dataset, we found that *eslec* and *itln1* are both highly expressed in eosinophils, while *itln1* is also weakly expressed in neutrophils (Fig. S9). Based on the stable expression in both homeostatic and pathological eosinophils, and larval zebrafish specificity, *eslec* is identified as a pan-eosinophil lineage marker for developing, steady-state, and activated eosinophils.

The spatial-temporal developmental pattern of eosinophils was firstly studied in the *Tg(eslec:eGFP)* line. From 5 dpf onwards, *eslec*⁺ eosinophils are gradually evident in the CHT region, the kidney, and intestine of zebrafish larvae. These locations are equivalent to the mammalian fetal organs (respectively, fetal liver, bone marrow, and intestine)[43]. Murine eosinophils are known to be found in these locations as well[44–46]. In adult zebrafish, eosinophils could be found in KM, IPEX, intestine, spleen, and liver, similar to the distributions in

humans[47]. Together, these data suggested a conserved eosinophilopoiesis pattern across species.

We previously showed that zebrafish Cebp1 is the functional orthologue of mammalian C/EBPε in regulating neutrophil development[36]. Herein, we demonstrated Cebp1 to suppress eosinophilic cell commitment and inhibit the differentiation and maturation of the eosinophil lineage. Mutation of the *CEBPE/Cebpe* gene exhibits severe eosinophil defects in both human[48,49] and mice[17], suggesting that mammalian C/EBPε may be positive regulators of eosinophilopoiesis. However, among the four isoforms of the human C/EBPε protein, mouse C/EBPε is only similar with human C/EBPεᵖ³² and C/EBPεᵖ³⁰, which are all proven to be required for eosinophilopoiesis[40,41]. The two repressor isoforms (C/EBPεᵖ²⁷ and C/EBPεᵖ¹⁴), of which mice lack their relevant orthologue, could inhibit the expression of eosinophil granule proteins and IL-5R[40]. C/EBPεᵖ²⁷ contains multiple repressor domains and inhibits MBP1 transcription, while C/EBPεᵖ¹⁴ contains only dimerization and DNA binding domains[40–42]. We overexpressed these mammalian C/EBPε isoforms and found that only human C/EBPεᵖ²⁷ inhibited eosinophilopoiesis in zebrafish, suggesting its similar transcriptional repressing role to zebrafish Cebp1. Taken together, it is suggested that zebrafish Cebp1 is the functional orthologue of human C/EBPεᵖ²⁷, but not other mammalian C/EBPε isoforms, in suppressing eosinophil development.

In both mammals and zebrafish, C/EBPβ is critical for emergency myelopoiesis[38,50] but dispensable for homeostatic neutrophil development[38,51]. Whether C/EBPβ affects the eosinophil lineage was unclear before. Early studies in chicken and human cell lines suggest that C/EBPβ induced eosinophil development by triggering lineage commitment and maturation[52,53]. Herein, we demonstrated, in vivo, that Cebpβ promotes the commitment and probably differentiation of eosinophils in zebrafish. For the molecular mechanisms, a previous study suggested that C/EBPβ cooperates with GATA-1 and transactivation of the MBP promoter in human HT93A cells[54]. In our study, we found that Cebpβ could bind to *eslec* promoter region and elevate the expression of *eslec*, which shares similarities with mammalian MBP, suggesting the conserved roles of C/EBPβ across different species. In addition, we also found zebrafish Cebpβ to be directly regulated by Cebp1. Thus, whether mammalian C/EBPβ controls eosinophilopoiesis or is similarly regulated by C/EBPε or other TFs is worth future investigation.

The function of zebrafish eosinophils remains largely unknown. ScRNA-Seq of the mature zebrafish eosinophils indicated that they highly express *cxcr4b*, *ccr9a*, *il1rl1*, and *fcer1gl*, which mammalian orthologues are associated with eosinophil behaviors[55–58]. The identified eosinophil marker, *eslec*, encodes a C-type lectin as the mammalian eosinophil major basic protein[28], suggesting putative similarities in granule proteins features and further functions across species[59]. Future studies focusing on functional assessment, disease models, and drug screening using the *Tg(eslec:eGFP)* line may lead to more exciting discoveries.

In summary, this study systemically identifies zebrafish eosinophil developmental patterns, putative functions, lineage landscapes, and genetic regulations. Given that the zebrafish model provides unique advantages for the study of hematopoiesis as well as blood- and immune-related diseases, the study of eosinophil biology in zebrafish will be a valuable complement to mammalian scientific discovery.

## Methods

### Animal maintenance

Zebrafish were maintained and bred based on *The Zebrafish Book*[60]. The zebrafish lines used in this research were: *Tg(gata2a:eGFP)*[21], *Tg(cdh17:DsRed)* (a gift from Dr. Hua Ruan of Southwest University, China), *Tg(lyz:DsRed)*[61], *Tg(mpeg1:loxP-DsRed-loxP-eGFP)*[32], *Tg(coro1a:loxP-DsRed-loxP-eGFP)*[30], *cebp1*ˢᵐᵘ¹ mutant[36], and *cebpb*ˢᶻʸ⁷ mutant[38]. All studies involving zebrafish were reviewed by the Animal Research

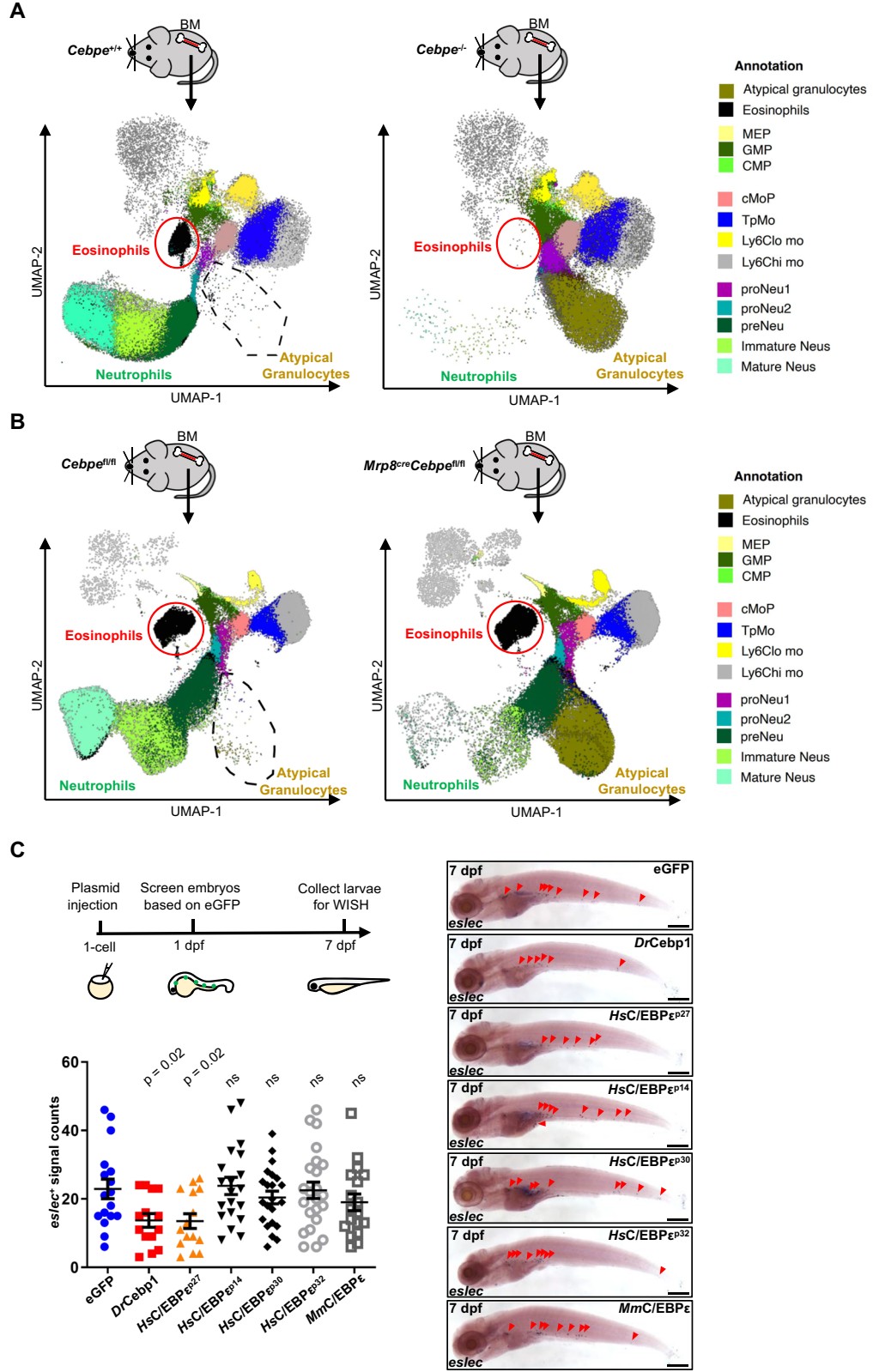

Advisory Committee of the South China University of Technology, Guangzhou, China.

Mice were bred and maintained under specific pathogen-free conditions (12 h light/12 h darkness, 22.5 °C, 52.5% humidity) in the Biological Resource Centre of A*STAR, Singapore. The *Mrp8^{cre}* (B6.Cg-Tg(S100A8-cre-eGFP)1Ilw/J) mice were obtained from the Jackson Laboratory and were used to cross with the *Cebpe^{fl/fl}* mice, generated through homologous recombination insertion of *Cebpe* gene containing loxP sites flanking exon 1 and 2. In the targeting vector, the Neo cassette is flanked by self-deletion sites and DTA was used for negative selection. After breeding, the conditional knockout *Mrp8^{cre}Cebpe^{fl/fl}* strain was verified by flow cytometry and genotyping. All studies involving mice were performed under the approval of the Institutional Animal Care and Use Committee, in accordance with the guidelines of

**Fig. 7 | Cebp1 is the orthologue of human C/EBPε[p27] in suppressing eosinophilopoiesis. A** Full *Cebpe* knockout affects both neutrophils and eosinophils in mice. BM cells of *Cebpe*[+/+] and *Cebpe*[−/−] mice were collected, stained with 15 different antibodies, and applied to flow cytometry analysis. The UMAPs were shown, representing the clustering information of mouse BM cells. Red circles marked the eosinophils, while the black-dotted regions marked the atypical granulocytes. The entire annotation information was shown on the right panel. **B** Neutrophil-specific *Cebpe* knockout induces atypical granulocytes in mice. BM cells of *Cebpe*[fl/fl] and *Mrp8*[cre]*Cebpe*[fl/fl] mice were collected, stained with 15 different antibodies, and applied to flow cytometry analysis. The UMAPs were shown, representing the clustering information of mouse BM cells. **C** Overexpression of human C/EBPε[p27]

mimicked Cebp1 phenotype. WT embryos were injected with *coro1a*-drived overexpression plasmids at 1-cell stage, then screened at 1 dpf based on eGFP signals, and finally collected at 7 dpf for WISH. Eosinophil (red arrowhead) numbers were decreased in larvae injected with *coro1a:cebp1-eGFP* (*Dr*Cebp1) or *coro1a:CEBPE*[p27]-*eGFP* (*Hs*C/EBPε[p27]) plasmid, while remained unchanged in other groups (bar = 200 μm). Quantification was performed using Student's *t* test (each group was only compared with *coro1a:eGFP* (eGFP) control, two-sided, mean ± SEM, ns represents no significance). Sample size: eGFP, *n* = 17; *Dr*Cebp1, *n* = 14; *Hs*C/EBPε[p27], *n* = 15; *Hs*C/EBPε[p14], *n* = 20; *Hs*C/EBPε[p30], *n* = 22; *Hs*C/EBPε[p32], *n* = 23; *Mm*C/EBPε, *n* = 18. Three independent experiments were conducted. Source data are provided as a Source Data file.

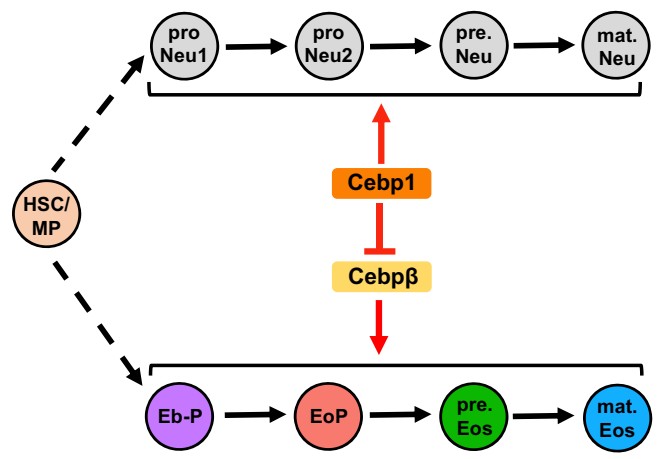

**Fig. 8 | The working model showing zebrafish Cebps in balancing granulopoiesis.** HSC or myeloid progenitor (MP) undergo stepwised differentiation for granulopoiesis. The neutrophil lineage cells contain proNeu1, proNeu2, pre.Neu, and mat.Neu, which nomenclature is based on a previous study[39]. The eosinophil lineage cells contain Eb-P, EoP, pre.Eos, and mat.Eos. For granulopoiesis, Cebp1 is essential for the generation of the neutrophil lineage, and Cebp1 suppresses Cebpβ to inhibit the commitment and differentiation of the eosinophil lineage.

the Agri-Food and Veterinary Authority and the National Advisory Committee for Laboratory Animal Research of Singapore.

### Generation of transgenic zebrafish lines

Transgenic lines were generated based on the Tol2 transposon system in AB background zebrafish[62]. To generate the *Tg(eslec:eGFP)* or *Tg(eslec:DsRed)* line, a 4.3 kb *eslec* promoter was used to trigger the expression of eGFP or DsRed in the pTol2 vector. For *Tg(hsp70:cebp1-eGFP)* or *Tg(hsp70:cebpb-eGFP)*, a 1.5 kb *hsp70* promoter was used to drive the overexpression of in-frame Cebp1-eGFP or Cebpβ-eGFP, thus generating fusion proteins for ChIP analysis. These vectors were co-injected into one-cell stage zebrafish embryos along with transposase mRNA. F0 larvae with ideal fluorescence patterns were raised and bred to generate the transgenic lines.

### Transient overexpression of mammalian C/EBPε proteins in zebrafish

A 7.0 kb *coro1a* promoter was used to induce the myeloid-specific overexpression of Cebp1-eGFP, mouse C/EBPε-eGFP, human C/EBPε[p32]-eGFP, human C/EBPε[p30]-eGFP, human C/EBPε[p27]-eGFP, and human C/EBPε[p14]-eGFP fusion proteins. The plasmids were separately injected into one-cell stage zebrafish embryos, in which those with strong eGFP fluorescence were collected at 7 dpf for WISH.

### WISH

WISH experiments were performed based on the standard protocol[63]. To obtain the probe for *eslec*, full-length *eslec* cDNA was ligated into the pBSK(+) vector. The antisense *eslec* mRNA was in vitro transcribed

by T7 RNA polymerase (ThermoFisher, USA) and labeled with Digoxigenin 10× RNA Labeling Mix (Roche, Switzerland) as the final probe. WISH signals were counted manually within each Z-plane. Finally, photographs of different Z-planes were merged by ZEN software (ZEISS, Germany).

### Cell morphology analysis

For the staining of *eslec*[+] cells, 7 dpf *Tg(eslec:eGFP)* larvae and adult *Tg(eslec:eGFP)* (3-month old, gender undistinguished) kidneys were prepared as cell suspensions with *eslec*[+] cells sorted by FACS based on eGFP fluorescence. The cell suspensions were applied to a Shandon Cytospin 4 (ThermoFisher) and stained with Giemsa (Sigma-Aldrich, Germany) and May-Grunwald's solution (Sigma-Aldrich).

### Confocal microscopy

To detect the developmental patterns of eosinophils, live *Tg(eslec:eGFP)* larvae were mounted in 1% low-melting agarose with 0.02% tricaine. The mounted larvae were assessed by confocal microscopy with a ZEISS LSM 800. To generate full-length photographs, entire larvae were visualized by tile scan with photographs merged based on different views. To detect colocalization of eosinophils and other cells, *Tg(eslec:eGFP)* was crossed with *Tg(cdh17:DsRed)*, *Tg(lyz:DsRed)*, *Tg(mpeg1:loxP-DsRed-loxP-eGFP)*, or *Tg(coro1a:loxP-DsRed-loxP-eGFP)*, and *Tg(eslec:DsRed)* was crossed with *Tg(gata2a:eGFP)*. The live larvae were assessed with a ZEISS LSM 800 at 5 or 7 dpf.

### Single-cell analysis

For scRNA-Seq analysis of eosinophils from KM, adult (3-month old) *Tg(eslec:eGFP)* kidneys were dissected from 5 individuals as a pool (gender undistinguished) and pipetted into cell suspensions in PBS with 0.04% BSA. Cell suspensions were stained with propidium iodide (PI), gated on eGFP[+]PI[−] cells, and sorted in this manner to obtain live KM eosinophils. The eosinophils were evaluated by trypan blue staining to confirm viability prior to 10x 3' (v3.1) scRNA-Seq gel beads-in-emulsion (GEMs) generation. Post GEM generation, library construction was performed based on the manufacturer's instructions (10x Genomics, USA). Sequencing was performed with a NovaSeq 6000 system (Illumina, USA). Mapping was performed by Cell Ranger (10x Genomics, 6.0.1, zebrafish genome: GRCz11). Expression matrices were analyzed with Seurat (3.2.3)[64]. Low-quality or contaminating cells were filtered based on total gene counts (nFeature > 200, final mean nFeature = 1,231.73), total reads (nCount > 1000, final mean nCount = 8,638.02), mitochondrial gene counts (percentage <15%), the expression level of *eslec* (percentage >0%), and doublet features. Finally, a total number of 2,092 eosinophil lineage cells passed the quality control thresholds. The cell cycle status of analyzed cells was assigned based on the expression of S phase genes and G2/M phase genes, as described previously[65]. The S phase and G2/M phase cells were both proliferating, but their transcriptomes varied greatly. Therefore, a Seurat partial cell cycle regression was used to regress differences between the S and G2/M phases, with signals separating non-proliferating and proliferating cells reserved. After final clustering, the specifically expressed genes of each cluster (Supplementary

Data 1) were analyzed by GO and KEGG enrichment using Metascape[66]. Trajectory analysis were performed with Monocle3 (0.2.3.0) and velocyto.R (0.6) (reads mapped to intronic regions = 14.9%, deltaT = 1, kCells = 40)[67,68].

In addition, the whole kidneys of adult WT, *cebp1*[−/−], and *cebpb*[−/−] zebrafish were dissected and pipette-mixed into cell suspensions. For each genotype, three 3-month-old female individuals were selected (*n* = 3), to avoid potential sex differences. As viability was >90% based on trypan blue staining, the cells were directly used for 10 × 3′ scRNA-Seq GEMs generation and further sequencing. Low-quality or contaminating cells were filtered based on total gene counts (nFeature > 200, final mean nFeature = 730.06), total reads (nCount > 1000, final mean nCount = 3,737.38), mitochondrial gene counts (percentage < 15%), and doublet features. For data consistency, the KM cells from three genotypes were integrated for analysis, but shown separately. The integration was performed with the "IntegrateData" function of Seurat, which utilized reciprocal principal component analysis to remove batch effects. Finally, a total number of 101,598 KM cells from different genotypes (WT-1: 8,262 cells, WT-2: 12,400 cells, WT-3: 13,095 cells, *cebp1*[−/−]−1: 12,218 cells, *cebp1*[−/−]−2: 12,384 cells, *cebp1*[−/−]−3: 10,555 cells, *cebpb*[−/−]−1: 14,440 cells, *cebpb*[−/−]−2: 10,326 cells, *cebpb*[−/−]−3: 7,918 cells) were analyzed. After partial cell cycle regression, the clusters were renamed based on their marker genes. The eosinophil count of each sample (WT-1: 124 cells, WT-2: 171 cells, WT-3: 89 cells, *cebp1*[−/−]−1: 319 cells, *cebp1*[−/−]−2: 364 cells, *cebp1*[−/−]−3: 332 cells, *cebpb*[−/−]−1: 37 cells, *cebpb*[−/−]−2: 43 cells, *cebpb*[−/−]−3: 25 cells) was divided by the total cell number of that sample to calculate the ratio. Post clustering, the eosinophil lineage cells were further subclustered based on the marker genes information, and the eosinophil count of each subcluster was divided by the total eosinophil number of the relevant sample to calculate the relative ratio. The marker genes and differentially expressed genes upon mutation are listed in Supplementary Data 6 (each cluster of all KM cells) and Supplementary Data 7 (each subcluster of eosinophil lineage cells).

## Bulk RNA-Seq

For all the bulk RNA-Seq analysis, we performed mini-bulk RNA-Seq with ~500 cells for each sample. To compare the transcriptomes of *cebp1*[+/+] and *cebp1*[−/−] eosinophils, 7 dpf *cebp1*[+/+];*Tg(eslec:eGFP)* and *cebp1*[−/−];*Tg(eslec:eGFP)* larvae were ground and digested with Dispase (ThermoFisher) into cell suspensions. To compare the transcriptomes of *cebpb*[+/+] and *cebpb*[−/−] eosinophils, 7 dpf *cebpb*[+/+];*Tg(eslec:eGFP)* and *cebpb*[−/−];*Tg(eslec:eGFP)* larvae were also prepared into cell suspensions. All cell suspensions were assayed by FACS and eGFP[+] cells sorted. Two replicates were generated for each sample, each collected from different individual pools. The subsequent library construction was modified based on a Smart-seq2 methodology[69], in which we enlarged the reaction volume and reduced the amplification cycle numbers, due to the much larger cell input (~500 cells) than the original report (single cell). Sequencing was performed with a NovaSeq 6000 system (Illumina). Mapping was performed by STAR[70] (2.7.10b, zebrafish genome: GRCz11). The mapped reads were annotated with featureCounts[71] (Rsubread, 2.0.1). Finally, the differentially expressed genes were analyzed with DESeq2[72] (1.26.0).

## ChIP-Seq

Because neither anti-Cebp1 nor anti-Cebpβ antibody is available, anti-GFP antibody (Abcam, UK, ab6658) was used to enrich DNA with interaction of in-frame fusion proteins, Cebp1-eGFP and Cebpβ-eGFP. We used *Tg(hsp70:cebp1-eGFP)* and *Tg(hsp70:cebpb-eGFP)* larvae to enrich these fusion proteins by activating the *hsp70* promoter at 39 °C. Larvae were heat-shocked for 2 hours/day from 2–5 dpf. At 5 dpf, ~200 larvae for each sample were lysed and crosslinked with 1% formaldehyde (ThermoFisher). The lysates were then sonicated with a Covaris M220 ultrasonicator (Duty Factor = 10%, peak incident

power = 75, Cycles of Burst = 200, Time = 10 min). The input samples were collected directly from sonication products (1% volume) as controls, while the rest of the products were incubated with anti-GFP antibodies (1:200). Then, Dynabeads Protein G (ThermoFisher) was added to enrich the antibody-DNA complex, which was de-crosslinked at 65 °C and the DNA was finally purified. Library construction was prepared with purified DNA using NEBnext Ultra II (New England Biolabs, USA), with sequencing by a NovaSeq 6000 system (Illumina). For data analysis, reads were aligned to the zebrafish genome (GRCz11) with BWA-MEM (0.7.17)[73]. Aligned reads were then filtered by Samtools (1.15.1)[74], with peak-calling by MACS2 (2.2.7.1)[75]. Peaks were annotated with ChIPseeker (1.22.1)[76] to identify Cebp1- and Cebpβ-binding genes. The aligned reads were converted from BAM formats into Bigwig formats by deepTools (3.5.1)[77]. Bigwig files were visualized with the Integrative Genomics Viewer[78].

## Dual-Luciferase assay

To verify the interaction of Cebp1 with the 3′UTR region of the *cebpb* gene, we cloned the 3′UTR region sequence of *cebpb* into the pMIR-Report (Ambion, USA) vector. For the overexpression of Cebp1, the entire coding sequence of *cebp1* was cloned and inserted into the pcDNA3.1 (Invitrogen) plasmid. HEK-293T cells were plated into 24-well plates, with each well transfected with 500 ng of pMIR-Report-*cebpb*-3′UTR or pMIR-Report (as control), 100 ng of pcDNA3.1-Cebp1 or pcDNA3.1 (as control), and 10 ng of pGL4.75[hRluc/CMV] (Promega, USA). The cells were lysed at 24 hours post-transfection and assessed by the Dual-Luciferase Reporter Assay System (Promega) for detection of Firefly and Renilla luciferase activities. Firefly/Renilla ratios were used to estimate the transcription activities affected by the Cebp1/*cebpb* interaction.

To verify whether the C/EBPs-eGFP fusions affect their TF activities, these fusion proteins were inserted into the pcDNA3.1 plasmid for transfection. The zebrafish Cebp1-eGFP vector was co-transfected with pMIR-Report-*cebpb*-3′UTR or pMIR-Report, as shown above. The human *Hs*C/EBPε-eGFP vectors were co-transfected with the pGL4-*HsMBP-P2*-Luc vector to examine whether it could affect the promoter activity of *HsMBP-P2*, which is the core promoter region of human *MBP* gene[41]. The mouse *Mm*C/EBPε-eGFP vector was also co-transfected with the pGL4-*MmMBP-P2*-Luc vector to examine its TF activity. pGL4.75[hRluc/CMV] vector was involved in all groups as a control and the subsequent analysis was listed as above.

## Flow Cytometry of zebrafish eosinophils

To evaluate the eosinophil percentages in larval and juvenile zebrafish, 4 dpf, 5 dpf, 7 dpf, 21 dpf, and 49 dpf *Tg(eslec:eGFP)* fish were ground and digested with Dispase (ThermoFisher). To detect the eosinophil distribution in adult fish organs, WT, *cebp1* mutant, or *cebpb* mutant *Tg(eslec:eGFP)* fish were sacrificed. The IPEX was collected as reported[21]. The kidney were collected and pipetted into cell suspensions. For other tissues except blood (spleen, intestine, brain, heart, and liver), organs were dissected and digested with Collagenase IV (Sigma). The cell suspensions were washed and applied to flow cytometry. The eosinophils were gated on eGFP[+]DsRed[−] to exclude autofluorescent cells. Finally, the flow cytometry data were analyzed by FlowJo.

## RT-qPCR

The eGFP[+] eosinophils were respectively sorted from adult *cebp1*[+/+] and *cebp1*[−/−] *Tg(eslec:eGFP)* KM (each sample from different individual). Then, the eosinophils were lysed with TRIzol reagent (ThermoFisher) and the RNA was extracted. The cDNA was generated with a HiScript II Q Select RT SuperMix kit (Vazyme, China). RT-qPCR was performed using on a LightCycler 96 system (Roche) with SYBR Green PCR Core Reagent kits (Roche). The RT-qPCR primers used were: *ef1a* (FP: GAGAAGTTCGAGAAGGAAGC; RP: CGTAGTATTTGCTGGTCTCG)

and *cebpb* (FP: TGCCCCAGTACCAGCATCTGG; RP: CGCTCGGTCAGCG AGATGTAGT). Finally, the relative expression of *cebpb* (fold changes) was analyzed by the ΔΔ comparative threshold method.

### Flow Cytometry of mouse BM cells

Analysis of mouse BM cells was performed as described previously[39]. Briefly, 6–10-weeks-old mouse (gender undistinguished) femurs were flushed using a 23–gauge needle in PBS containing 2 mM EDTA and 2% fetal bovine serum and passed through a 70–mm nylon mesh sieve. For the identification of BM myeloid progenitor cell subsets, cells were stained with fluorophore-conjugated anti-mouse antibodies against CD34 (RAM34, ThermoFisher, 13-0341-81, 1:100), CD11b (M1/70, ThermoFisher, 63-0112-80, 1:400), CD16/32 (2.4G2, BD Horizon, 565502, 1:400), CD115 (AFS598, Biolegend, 135510, 1:400), cKit (2B8, ThermoFisher, 62-1171-82, 1:400), CD101 (BB27, Biolegend, 331007, 1:400), CXCR4 (2B11, ThermoFisher, 13-9991-82, 1:400), Gr1 (RB6-8C5, ThermoFisher, 45-5931-80, 1:800), Ly6C (HK1.4, Biolegend, 128026, 1:400), SiglecF (E50-2440, BD Biosciences, 562757, 1:400), and CD106 (429, Biolegend, 105716, 1:400), together with exclusion lineage markers that include Ly6G (1A8, Biolegend, 127618, 1:200), CD90.2 (53-2.1, Biolegend, 140314, 1:400), B220 (RA3-6B2, ThermoFisher, 13-0452-82, 1:400) and NK.1.1 (PK136, ThermoFisher, 13-5951-81, 1:400). To visualize *Cebpe^{fl/fl}* and *Mrp8^{cre}Cebpe^{fl/fl}* BM cells, UMAP analysis was performed using the in-built FlowJo UMAP plugin.

### Statistical analysis

Statistical analysis between two groups were performed using Student's *t* test. In these analyses, $P < 0.05$ was considered statistically significant. All analyses were performed in GraphPad Prism 8.0.2.

### Reporting summary

Further information on research design is available in the Nature Portfolio Reporting Summary linked to this article.

## Data availability

All sequencing data generated in this study has been uploaded to the NIH Gene Expression Omnibus under accession number GSE198314. The zebrafish genome version used in the present study is GRCz11. Source data are provided in this paper.

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

## Acknowledgements

We thank Drs. David Traver, Hua Ruan, and Honghui Huang for sharing the *Tg(gata2a:eGFP)* and *Tg(cdh17:DsRed)* transgenic lines with us. We thank Dr. Jianbin Wang for the methodology of mini-bulk RNA-Seq. We thank Drs. Kuangyu Yen, Jun Hou, Bowei Han, Junfeng Huang, and Mr. Houyu Zhang for their help in the bioinformatics analysis. We thank Drs. Zilong Wen, Wenqing Zhang, and Jin Xu for their suggestions and shared materials. We thank all the staff at the research platform and zebrafish facility of SCUT and the High-Throughput Analysis Platform of Sun Yat-Sen University Cancer Center. This work was supported by the National Key R&D Program of China (2023YFA1800100 and 2018YFA0800200, both received by Y.Z.), the National Natural Science Foundation of China (31922023, received by Y.Z., 32100664, received by G.L., and 32100665, received by M.W.), the Guangdong Province Universities and Colleges Pearl River Scholar Funded Scheme (2019, received by Y.Z.), and the Guangzhou Municipal Science and Technology Project (202201010176, received by J.L.).

## Author contributions

Y.Z. supervised this study; G.L. and Y.S. performed most experiments and data analysis; G.L. and Y.Z. wrote the manuscript; I.K. and L.N. helped with the mouse flow cytometric analysis, data comparison, and revised the manuscript; L.Y., W.W., and P.H. helped with the developmental pattern experiments; M.W. and J.L. helped with the marker (*eslec*) identification; Z.H. helped with the Cebp1-related research; Z.L. and F.G. helped with the *Cebpe* conditional knockout mouse data; S.H., W.P., and J.B. helped with the scRNA-Seq analysis.

## Competing interests

The authors declare no competing interest.
