## [Peer Review File · Nature Communications]

Cebp1 and Cebp β transcriptional axis controls eosinophilopoiesis in zebrafishREVIEWER COMMENTS

Reviewer #1 (Remarks to the Author):

Summary paper:

In this manuscript, Gaofei Li, Yicong Sun et al. identified, generated and characterized a novel reporter line (Tg(eslec:eGFP)) for the labelling of eosinophils from early life to adulthood. With a combination of computational analyses and experimental validations, they unraveled a transcriptional interplay between Cebp1 and Cebpb that is relevant for eosinophils lineage commitment and maturation. Specifically, the authors observed that Cebp1 negatively regulated eosinophils specification via Cebpb repression. Lastly, they linked Cebp1 and Cebpb observations in mammals. However, major points and inconsistencies should be addressed.

Major comments:

- The authors showed the pattern of expression of their newly established Tg(eslec:eGFP) transgenic line; however, they mainly focused on the early developmental stages. Can the authors investigate and show the localization of eslec+ cells in adulthood?
- Can the authors validate all the WISH and imaging analysis in this manuscript (Fig. 1B, 1C; Fig. 3A, 3B; Fig. 5A, 5B; Fig. 7C; Fig. S1E) by checking the frequency of eslec+ cells with FACS?
- Can the authors experimentally validate the cell cycle analysis shown in Fig. 2C?
- The authors claimed that the cell cycle analysis (scRNA-seq dataset, Fig. 2C) shows a high percentage of proliferating cells (line 137-139). However, only 34% of these are in S/G2/M phase. Please argument on this.
- Can the authors discuss why, in the RNA-seq experiments (Fig. 4, Fig. 6), they decided to use Tg(gata2a:eGFP) instead of their newly generated Tg(eslec:eGFP) transgenic line? Could they perform the same experiment using their more specific and lineage stringent transgenic line?
- Can the authors discuss their choice of performing RNA-seq analyses exploiting adult zebrafish, and ChIP-seq analyses in zebrafish larvae (Fig. 4, Fig. 6)? Could they eventually perform these analyses at the same age and possibly considering the same cell population?
- In Fig. 4E, the authors showed the result of the Luciferase Assay on HEK-293T cells, to prove that cebp1 negatively regulates cebpb via direct interaction with cebpb 3'UTR. Can the authors prove this by showing cebpb expression levels in eslec+ cell population in absence or presence of cebp1?
- The authors showed that cebp1^{-/-} larvae have reduced number of eslec+ cells and that cebp1^{-/-} adults seem to have impaired eosinophilopoiesis. Can the authors show if Eos maturation is impaired in cebp1^{-/-}? Or is cebp1 negatively regulating only the Eosinophil lineage commitment?
- Can the authors investigate if cebp1 manipulation affects eosinophilopoiesis also in adult zebrafish, and if it affects also the frequency of peripheral Eos?
- Can the authors show the effect of lack of cebpb in adult zebrafish as well? How does the Eos frequency changes within the KM and peripherically?
- Lack of cebpb seems to drastically affect not only Eosinophils, but also other cell populations (Fig. 5C). Can the authors discuss this?
-

Minor comments:

- In Fig. 2F, the axis labels are missing. Could the authors add them please?
- Can the authors explain how they have calculated the percentages of Eos population and subpopulations in Fig. 3C, 3E, 5C, and 5D?
- Can the authors explicitly describe which material and methods have been used for the preparation of RNA-seq libraries and for the subsequent analysis?
- Can the authors explicitly describe which material and methods have been used for the preparation of ChIP-seq libraries?
- Can the authors state in the Figure Legends how many independent experiments have been performed and the number of biological replicates per experimental group?
- Please revise word choice in the manuscript (e.g. line 85, 252).

- To make it easier for the reader to understand, the authors should introduce *cebpb* as soon as they name it. Please move Line 225-227 at the end of Line 218.
- Please revise the discussion of the manuscript, in order to make it easier to read and follow.

Reviewer #2 (Remarks to the Author):

Li et al have identified "eslec" as a gene expressed in zebrafish eosinophils and have use the eslec promoter to greater reporter lines. Descriptive studies document the anatomical distribution of eslec-marked cells in early zebrafish development. Single cell transcriptomics identify a developmental trajectory for zebrafish eosinophils. Genetic studies are used to explore the regulation of eosinophil development by transcription factors, particularly of the C/EBP family. The major value of the paper is the utility it adds to the zebrafish model for future studies of developmental, physiological and disease processes involving eosinophils.

Comments

1. Co-localisation data of eslec:eGFP reporter gene expression with other reporter genes. The data in Fig S1G, and the interpretations made about reporter gene specificity are based only on the handful of cells in these selected fields. This is anecdotal evidence. A rigorous analysis to underpin these and future studies requires bigger denominators and quantitative scoring: at the very least, more cells from more fields, and preferably from multiple fish. I suggest that an appropriately powered experiment providing a point estimate of specificity in both analytical directions would score at least $n=100$ randomly-selected eslec +ve cells for expression of the other reporter, and at least $n=100$ DsRed cells of the comparator reporter for eslec positivity. It would also be of interest to know the abundance of *lyz*, *mpeg1* and *coro1a* transcripts in the eslec+ve scRNAseq population (which likely will reveal that transcript expression is less lineage-confined than the reporter genes alone suggest).

2. Figure 3C and associated text. The data are interpreted to indicate over-representation of the "integrated eosinophil lineage", from 1.6 to 3%. This however is a proportional over-representation of this lineage in the population of approx. 500 input cells that were analysed, and not necessarily indicative of an absolute over-abundance. A whole major population of cell types (neutrophils) is largely absent – that alone will increase the proportional representation of the remaining lineages. Then there is the appearance of an expanded population – the abnormal eslec-expressing described in lines 176-9; there is an impact of this to reduce the proportional representation of the remaining lineages. Finally, without knowing the total cellularity of the KM, just what these proportional changes mean for productivity and abundance in the whole population that is being represented is unknown. The data either need to address the relationship of proportional representation to absolute population sizes, or present appropriate caveats.

A way around this would have been to cross the eslec:eGFP reporter onto the two mutant backgrounds and gate on the eslec+ population, as is done for WT in Figure 2. However, it is recognised that this is a 2-generation experiment (unless the transgene were directly introduced by injection into homozygous mutant embryos, but this would introduce potential for independent reporter line specific idiosyncrasies).

The same considerations apply to the comparative comments made about the *cebpb* mutant data in Figure 5C, although I note that a 5-fold change in proportionality is much greater than the 2-fold change in the *cebpb1* mutant.

3. Robustness of observations made on subsetting small datasets when there is only $n=1$ dataset. Figs 3E and 5D represent 4-category comparative maturation profiles of subpopulations of cells that amount to 0.3-3% of the total population of cells. However, it is easy to lose track of the actual n

values for minor populations in complex scRNAseq datasets, which always appear large and data rich. The Methods section (line 453) suggests that there are approximately 11,000 cells/genotype (34K cells in total for 3 genotypes). Calculating therefore: 0.3% of 11K cells is 33 cells, and 16.2% of 33 cells (the smallest subset) is n=5 cells. Even though these categorical maturation profile data are presented in merely descriptive terms, and no analytical stats are performed, in any other setting, this would be considered too underpowered and under-replicated for the comparison to be of value. An ordered categorical comparative analysis could be performed (but with these low numbers per category, it is likely to be too underpowered to confirm that is a difference beyond what could be expected by chance alone).

4. Replication and validation of key functional experiments. Figure 7C bases the demonstrated of a shared repressive functional capacity for zebrafish *Cebp1* and human *C/EBP[epsilon]p27* on approx. n=15 embryos in these two test groups, with no indication of the degree of experimental replication. Indicate the degree of experimental replication. A reasonable expectation is that this experiment would be run at least n=3 times; n=15 embryos in each test group would be a reasonable minimum expectation the minimum given the variance seen here.

In this experiment, multiple *C/EBP* species had no biological effect, although their expression was validated by detecting EGFP expression from a carboxyl-fusion protein. The selection of *C/EBP[epsilon]p27* as the ortholog with exclusive shared functional capability relies on these being true negative results. For these no-effect species, what is the positive control that proves the carboxyl EGFP-fusion did not (or does not) inactivate the *C/EBP* transcription factor to which it was fused?

5. Line 350 acknowledges that "The function of zebrafish eosinophils remains largely unknown." Li et al do reference several prior studies providing some evidence for shared functionalities between mammalian and zebrafish eosinophils. However the term "eosinophils" refers to a cell that has an affinity for eosin and other acidic stains in each species, not to a cells that even look alike and which have very different nuclei and granule morphologies. Assigning abundant expression of a MBP (major basic protein) related protein to this leukocyte lineage in zebrafish seems to me to be an important contribution from this report that supports the use of a common name that implies a common identity and function. I suggest the authors consider strengthening this point in their discussion.

Minor points

1. Line 94-5. The insights for eosinophil-related diseases are potential future insights, not ones demonstrated in this paper (which did not model any specific disease), as is claimed here.
2. Lines 298-91 are not strictly a result – better a discussion point?
3. Line 292. "functional" not "function"

Reviewer #3 (Remarks to the Author):

In this study, the authors elucidate the development of a key immune cell type, Eosinophils, in zebrafish. While some studies of adult zebrafish have characterized mature Eosinophils, the ontogeny of these cells is poorly understood. The authors apply a wide range of approaches to dissect the developmental trajectory of these cells and identify two key regulators, *cebp1* and *cebpb*, which they demonstrate to be critical for eosinophil development. The authors go on to explore the conservation of these regulators in mice, and they identify both shared and disparate cell type-specific outcomes of gain and loss of function experiments. Overall, the authors perform a thorough characterization of eosinophil development in zebrafish and address the conservation of this process in mice. The integration of numerous experimental approaches is impressive, and the key points of each towards the main goal of the study provides a clear, logical through line. However, there are missing aspects of some important analyses that need to be addressed to facilitate proper interpretation of the data. And

some information is lacking that would make the study interesting and interpretable for a broader audience.

Major issues:

1. The authors surveyed adult tissue with the goal of describing eosinophil development, but it's not clear whether they expected to see cells at every developmental stage. Can the authors please clarify why they chose adult cells to explore the full developmental trajectory of these cells? Is there evidence that these cells are continuously developing over the lifespan of the animal?

2. The authors should report additional quality control information for their single cell RNA-sequencing experiments, such as UMIs per cell type, quality control cutoffs, and numbers of genes expressed. Additionally, the methods section should be expanded to include information about the 10X kit version that was used, as well as the versions of all software packages (Seurat, Cell Ranger) that were used to analyze the data.

3. From the current RNA velocity plot, it is difficult to visually summarize the directionality of the arrows. For instance, in the presumptive EoP's, it could be equally suggested by these data that the differentiation path is directly from EoP to mature eosinophil, rather than passing through the pre.Eos state. If the authors wish to support their Eosinophil trajectory inference using RNA velocity, it would be helpful to improve visibility of directionality in the velocity plot by adjusting the plotting parameters. Additionally, RNA velocity estimation relies heavily on the detection of spliced vs. unspliced mRNAs, and these can often vary from experiment to experiment or across different cell types. Can the authors please report these aspects of their input data in addition to an expanded methods section for any non-default parameters for the RNA velocity pipeline that they used (or note default and the version if that was the case).

4. The population of "atypical erythrocytes" in the *cebpb1*^{-/-} mutants is quite interesting, especially given the differences in zebrafish vs. mammalian erythrocytes (such as having a nucleus or not). To enhance the resource of this dataset and to provide additional information about this population of cells, it would be helpful for the authors to supply 1) an expanded list of cell type specific genes according to their annotations, and 2) a list of differentially expressed genes for each cell type for *cebpb1*^{-/-} and *cebpb*^{-/-} fish compared to wildtype.

5. The authors note that adult zebrafish were collected and pooled for the scRNA-seq experiments. However, since eosinophils are an immune cell type that is responsive to environmental factors (such as infections), can the authors report the age of the animals and whether the genotypes that were compared came from the same clutch of siblings and collected when the animals were the same age? Additionally, if it was possible to sex the animals (i.e. if they were old enough), the authors should report whether females or males or a mixture were used for their experiments. All of this information is important to understand and interpret the RNA-seq data.

6. In Figure 7, the authors report that mouse BM cells were subjected to InfinityFlow analysis, but that method is not described anywhere in the manuscript. Especially given that UMAP dimensionality reduction is used to display the single cell RNA-seq data, the authors should add clarification to the text, methods, and figure to make it clear that a different data type is being displayed. The authors should additionally list the antibodies or panel that was used or include a citation that lists them.

7. The authors have made the data publicly available in GEO. However, for the scRNA-seq data, only the count matrix is available as a processed data file. To enable reproducible analysis, the authors should also upload their gene metadata and cell metadata (including cell type annotations). Additionally, deposition of the Seurat objects would be extremely helpful for enabling others to benefit from the large amount of data produced in their study (similarly to their re-analysis of the Tang, et al. data included in Supp Fig. 1).

Minor issues:

1. To identify an eosinophil marker that is expressed early in development, the authors use a published single cell RNA-seq dataset of kidney cells. However, they 1) do not explain why the kidney is the proper place to look for these cells or 2) why the *gata2+* cluster would be the eosinophils. Citations and explanation of this key background information should be added. (line 100-104)
2. The authors indicate colocalization with *coro1a+* suggests eosinophil identity, but they do not explain why. A citation or small explanation should be added here.
3. "KM" eosinophils is an acronym that's used throughout the manuscript but never defined.
4. There is a wide range of font sizes throughout the figures, including some that are illegible (i.e. axis labels in Fig 3C). These should be improved for readability.
5. Numbers of cells should be noted wherever possible. For instance, no cell numbers are reported for the comparative analysis in Figure 3. Also, in line 453, it is not clear whether the ~34k cells is referring to the combination of the three genotypes or a subset of them.
6. What version of the zebrafish genome and transcriptome were used to map the reads for all experiments? These should be added to the analysis portion of the methods.
7. When the authors explain that they used RNA-seq in line 249, they should clarify that it is "bulk" RNA-seq to avoid confusion from the other single cell RNA-seq data in the paper and to highlight that this is a new experiment.
8. In line 251, the authors note that they're comparing RNA-seq data from adult, sorted eosinophils to ChIP-seq data from 7-dpf whole larvae. While I understand that this is likely due to experimental constraints on both the input requirements for ChIP-seq vs. RNA-seq as well as the challenge of using HS over-expression in adults, the authors should at least discuss this striking difference in age and tissue type and what implications it might have on their results.
9. For the UMAPs in figure 7, it would be very helpful to have direct labels for key cell populations rather than just the legend. Additionally, axes (or a representation of them) should be added to the UMAP plots.

Typos:

1. Line 211-213 is confusing and should be clarified.
2. There are a number of minor but distracting grammatical errors throughout the manuscript. An additional editing round to correct these would greatly improve overall readability.

Reviewer #1 (Remarks to the Author):

Summary paper:

In this manuscript, Gaofei Li, Yicong Sun et al. identified, generated and characterized a novel reporter line (Tg(*eslec*:eGFP)) for the labelling of eosinophils from early life to adulthood. With a combination of computational analyses and experimental validations, they unraveled a transcriptional interplay between *Cebp1* and *Cebpb* that is relevant for eosinophils lineage commitment and maturation. Specifically, the authors observed that *Cebp1* negatively regulated eosinophils specification via *Cebpb* repression. Lastly, they linked *Cebp1* and *Cebpb* observations in mammals. However, major points and inconsistencies should be addressed.

Major comments:

1. The authors showed the pattern of expression of their newly established Tg(*eslec*:eGFP) transgenic line; however, they mainly focused on the early developmental stages. Can the authors investigate and show the localization of *eslec*⁺ cells in adulthood?

Response: Thank you for your advice. Accordingly, we performed flow cytometry to detect the *eslec*⁺ eosinophils from different tissues in adulthood. Eosinophils were found in the KM, IPEX, intestine, spleen, and liver, but few in the brain, heart, and peripheral blood (new Fig. S2C-D). (Methods part, Page 20, Line 554-563, Results part, Page 5-6, Line 135-139, and Discussed in Page 12, Line 336-337)

2. Can the authors validate all the WISH and imaging analysis in this manuscript (Fig. 1B, 1C; Fig. 3A, 3B; Fig. 5A, 5B; Fig. 7C; Fig. S1E) by checking the frequency of *eslec*⁺ cells with FACS?

Response: Thank you for your advice. As suggested, we applied flow cytometry to validate Fig. 1B, 1C, and Fig. S1E and found that eosinophils were indeed absent in 4 dpf larvae, while emerged since 5 dpf and significantly increased at 7 dpf (new Fig. S2A-B). The flow cytometry results (new Fig. S2A-B) were in accordance with our previous quantification data from WISH and imaging experiments (previous Fig. 1B-C and Fig. S1E), so we think that the previous quantifications of WISH and imaging are also reliable (Fig. 3A, 3B; Fig. 5A, 5B; Fig. 7C).

3. Can the authors experimentally validate the cell cycle analysis shown in Fig. 2C?

Response: Thank you for your valuable advice. As suggested, we checked literatures for approaches to the validation. We found that researchers mainly applied flow cytometry assays to validate the proliferation features, such as based on the Fucci system, Mki67 expression, or DNA ploidy, but mostly on a qualitative level (Hanjie Li et al., *Cell*, 2019, Immanuel Kwok et al., *Immunity*, 2020). Here, we used Hoechst staining to label the DNA of KM *eslec*⁺ cells and performed flow cytometry to detect the cell cycle status of KM eosinophilic cells. The results exhibited that there indeed existed the proliferating cell population, which validates the scRNA cell cycle analysis. However, the rate of proliferating cells based on flow cytometry was ~13% (representative figure shown below, but not included in the revision), lower than that from scRNA-Seq analysis, which was probably due to the different approaches (DNA ploidy in Hoechst

assay, mRNA expression of S/G2/M-related genes in scRNA-Seq analysis). For this reason, we didn't show the Hoechst data in the revision, which might be a distraction. Nevertheless, there are proliferative EoPs in the eosinophil lineage cells from KM that could indeed be validated.

4. The authors claimed that the cell cycle analysis (scRNA-seq dataset, Fig. 2C) shows a high percentage of proliferating cells (line 137-139). However, only 34% of these are in S/G2/M phase. Please argument on this.

Response: We're grateful for your correction. We've modified this description in our revised manuscript (Page 6, Line 150-152), to avoid overstatement.

5. Can the authors discuss why, in the RNA-seq experiments (Fig. 4, Fig. 6), they decided to use Tg(*gata2a*:eGFP) instead of their newly generated Tg(*eslec*:eGFP) transgenic line? Could they perform the same experiment using their more specific and lineage stringent transgenic line?

Response: Thank you for your advice. In the revision, we performed a new bulk RNA-Seq with the eosinophils sorted from the Tg(*eslec*:eGFP) transgenic line (new Fig. 4B and Fig. S6A). The new sequencing data also showed *cebpb* upregulation in *cebpl* mutants (new Fig. 4C), which is in line with our previous data. For Fig. 6, we also reperformed a new bulk RNA-Seq with the Tg(*eslec*:eGFP) larvae as you suggested (new Fig. 6B and Fig. S6C). The results also suggested that *cebpb* affects eosinophil lineage differentiation (new Fig. 6), which is similar to our previous submission.

6. Can the authors discuss their choice of performing RNA-seq analyses exploiting adult zebrafish, and ChIP-seq analyses in zebrafish larvae (Fig. 4, Fig. 6)? Could they eventually perform these analyses at the same age and possibly considering the same cell population?

Response: Thank you for your suggestion. As suggested, we re-performed the RNA-Seq assay at the larval stage, which is in accordance with the stages of ChIP-Seq experiments (Fig. 4 and 6). (Methods part, Page 18, Line 496-500, and Results part, Page 8, Line 220; Page 10, Line 266)

We agree with the reviewer that performing sequencing with the same cell population is better. However, sorted eosinophils could not reach enough cell amounts for ChIP-Seq, so it is not

feasible in our hands. Therefore, we utilized the whole larvae to perform ChIP-Seq assays alternatively.

7. In Fig. 4E, the authors showed the result of the Luciferase Assay on HEK-293T cells, to prove that *cebp1* negatively regulates *cebpb* via direct interaction with *cebpb* 3'UTR. Can the authors prove this by showing *cebpb* expression levels in *eslec*⁺ cell population in absence or presence of *cebp1*?

Response: Thank you for your advice. Our bulk RNA-Seq results showed that *cebpb* was upregulated in *cebp1* mutants, compared with WT control (new Fig. 4C). As suggested, we also performed a Q-PCR assay with sorted eosinophils and found that *cebpb* was upregulated upon *cebp1* mutation (new Fig. 4D). (Methods part, Page 20-21, Line 565-575, and Results part, Page 9, Line 232-233)

8. The authors showed that *cebp1*^{-/-} larvae have reduced number of *eslec*⁺ cells and that *cebp1*^{-/-} adults seem to have impaired eosinophilopoiesis. Can the authors show if Eos maturation is impaired in *cebp1*^{-/-}? Or is *cebp1* negatively regulating only the Eosinophil lineage commitment?

Response: Thank you for your advice. Our results showed that *cebp1* mutants have an increased number of *eslec*⁺ cells in both larval and adult stages (Fig. 3A, 3C, and S4). Through scRNA-Seq, we further investigated the lineage changes upon *cebp1* mutation. We found that Eb-Ps, the eosinophil-lineage biased progenitors, were significantly increased in *cebp1* mutants (new Fig. 3E), indicating that eosinophil cell fate was promoted in *cebp1* mutants. On the other hand, the eosinophil precursors (pre.Eos) were slightly fewer in mutants (new Fig. 3E), suggesting that *cebp1* might also affect eosinophil differentiation. (Results part, Page 8, Line 202-207)

9. Can the authors investigate if *cebp1* manipulation affects eosinophilopoiesis also in adult zebrafish, and if it affects also the frequency of peripheral Eos?

Response: Thank you for your advice. In the revision, we performed flow cytometry to detect the eosinophil percentages in various adult tissues of both WT and *cebp1* mutant fish. The results showed that the eosinophil percentages were higher in the KM, IPEX, intestine, spleen, and liver in *cebp1* mutants (new Fig. S4). (Methods part, Page 20, Line 557-563, and Results part, Page 7, Line 183-185)

10. Can the authors show the effect of lack of *cebpb* in adult zebrafish as well? How does the Eos frequency changes within the KM and peripherically?

Response: Thank you for your advice. In the revision, we performed flow cytometry to detect the eosinophil percentages in various adult tissues of both WT and *cebpb* mutant fish. The results showed that the eosinophil percentages were lower in the KM, IPEX, intestine, spleen, and liver in *cebp1* mutants (new Fig. S7). (Methods part, Page 20, Line 557-563, and Results part, Page 9, Line 248-249)

11. Lack of cebpb seems to drastically affect not only Eosinophils, but also other cell populations (Fig. 5C). Can the authors discuss this?

Response: Thank you for your question. As another reviewer suggested that the n=1 dataset was too underpowered, we've reperfomed scRNA-Seq analysis with n=3 samples per genotype. Based on the new result with triplicated samples, only eosinophils were significantly reduced, while none of other cell populations was significantly affected (new Fig. 5C).

Minor comments:

1. In Fig. 2F, the axis labels are missing. Could the authors add them please?

Response: Thank you for your advice. We've added the labels in our revised manuscript.

2. Can the authors explain how they have calculated the percentages of Eos population and subpopulations in Fig. 3C, 3E, 5C, and 5D?

Response: Thank you for your advice. We've added the related descriptions in the Materials and Methods section (Page 17-18, Line 484-492).

3. Can the authors explicitly describe which material and methods have been used for the preparation of RNA-seq libraries and for the subsequent analysis?

Response: Thank you for your advice. We accordingly described this part in detail in the revision (Method part Page 18, Line 502-509, and new reference 68).

4. Can the authors explicitly describe which material and methods have been used for the preparation of ChIP-seq libraries?

Response: Thank you for your advice. We explicitly introduced our sample preparation in the revision (Page 19, Line 516-523).

5. Can the authors state in the Figure Legends how many independent experiments have been performed and the number of biological replicates per experimental group?

Response: Thanks for your comments. We've added the replication and sample size information in the revised figure legends. (Page 31, Line 859-860; Page 32, Line 883-884 and 887-888; Page 33, Line 909-910, 916-917, 922-923, and 926-927; Page 34-35, Line 965-966; Supplementary Figure Legends for Fig. S2, S4, S7, and S8)

6. Please revise word choice in the manuscript (e.g. line 85, 252).

Response: Thanks for your correction. We've rewritten the sentence in Page 4, Line 84-86 and changed the word "poorly expressed" into "downregulated" in Page 10, Line 268.

7. To make it easier for the reader to understand, the authors should introduce *cebpb* as soon as they name it. Please move Line 225-227 at the end of Line 218.

Response: Thanks for your correction. We've modified the related descriptions as you suggested (Page 9, Line 233-235).

8. Please revise the discussion of the manuscript, in order to make it easier to read and follow.

Response: Thanks for your advice. We've revised the discussion part (Page 12, Line 336-337; Page 13, Line 363-366; Page 14, Line 373-375) as you suggested.

Reviewer #2 (Remarks to the Author):

Li et al have identified “*eslec*” as a gene expressed in zebrafish eosinophils and have use the *eslec* promoter to greater reporter lines. Descriptive studies document the anatomical distribution of *eslec*-marked cells in early zebrafish development. Single cell transcriptomics identify a developmental trajectory for zebrafish eosinophils. Genetic studies are used to explore the regulation of eosinophil development by transcription factors, particularly of the C/EBP family. The major value of the paper is the utility it adds to the zebrafish model for future studies of developmental, physiological and disease processes involving eosinophils.

Comments

1. Co-localisation data of *eslec*:eGFP reporter gene expression with other reporter genes. The data in Fig S1G, and the interpretations made about reporter gene specificity are based only on the handful of cells in these selected fields. This is anecdotal evidence. A rigorous analysis to underpin these and future studies requires bigger denominators and quantitative scoring: at the very least, more cells from more fields, and preferably from multiple fish. I suggest that an appropriately powered experiment providing a point estimate of specificity in both analytical directions would score at least $n=100$ randomly-selected *eslec* +ve cells for expression of the other reporter, and at least $n=100$ DsRed cells of the comparator reporter for *eslec* positivity. It would also be of interest to know the abundance of *lyz*, *mpeg1* and *coro1a* transcripts in the *eslec*+ve scRNAseq population (which likely will reveal that transcript expression is less lineage-confined than the reporter genes alone suggest).

Response: Thank you for your valuable advice. As you suggested, we randomly selected *eslec*⁺ cells ($n > 100$) and checked whether they were also DsRed⁺ (new Fig. S1G). The percentage of eGFP⁺ cells in total *lyz*⁺, *mpeg1*⁺, and *coro1a*⁺ cells ($n > 100$) were also analyzed (new Fig. S1G). The results showed that the *eslec*:eGFP⁺ cells belonged to *coro1a*⁺ cells, but were not colocalized with *lyz*⁺ neutrophils or *mpeg1*⁺ macrophages (new Fig. S1G), confirming the lineage specificity of *eslec*. Besides, we showed the expression pattern of *lyz*, *mpeg1*, and *coro1a* in our scRNA-Seq data of KM eosinophils (new Fig. S3A), which results were similar to the confocal imaging data (new Fig. S1G). (Results part, Page 6, Line 148-150)

2. Figure 3C and associated text. The data are interpreted to indicate over-representation of the “integrated eosinophil lineage”, from 1.6 to 3%. This however is a proportional over-representation of this lineage in the population of approx. 500 input cells that were analysed, and not necessarily indicative of an absolute over-abundance. A whole major population of cell types (neutrophils) is largely absent – that alone will increase the proportional representation of the remaining lineages. Then there is the appearance of an expanded population – the abnormal *eslec*-expressing described in lines 176-9; there is an impact of this to reduce the proportional representation of the remaining lineages. Finally, without knowing the total cellularity of the KM, just what these proportional changes mean for productivity and abundance in the whole population that is being represented is unknown. The data either need to address the relationship of proportional representation to absolute population sizes, or present appropriate caveats.

A way around this would have been to cross the *eslec:eGFP* reporter onto the two mutant backgrounds and gate on the *eslec+* population, as is done for WT in Figure 2. However, it is recognised that this is a 2-generation experiment (unless the transgene were directly introduced by injection into homozygous mutant embryos, but this would introduce potential for independent reporter line specific idiosyncrasies).

The same considerations apply to the comparative comments made about the *cebpb* mutant data in Figure 5C, although I note that a 5-fold change in proportionality is much greater than the 2-fold change in the *cebpl* mutant.

Response: Thank you for your valuable advice. For your concerns, we think that the total cell amount didn't change drastically in *cebpl* mutants, because the emergence of atypical cells (~12%) was comparable to the absence of neutrophils (~18%) (new Fig. 3C). In our new triplicated scRNA-Seq data, we compared the eosinophil percentages in all KM cells and found them significantly increased in mutants (new Fig. 3C and Fig. S4C). We understand that using mutant *Tg(eslec:eGFP)* to examine the absolute population size of eosinophils may be helpful, while it is difficult to normalize in our hands. Alternatively, we applied the unaltered HSPC population (new Fig. S4D) to normalize the eosinophil percentages, and found the relative eosinophil/HSPC ratio also significantly increased in *cebpl* mutants (new Fig. S4E). Therefore, we concluded that the increase in eosinophilopoiesis upon *cebpl* mutation is reliable. (Results part, Page 7-8, Line 191-196)

For the *cebpb* mutants, we also performed a similar analysis which showed the decrease of eosinophil percentages (new Fig. 5C and Fig. S4C-E). In addition, no other cluster percentage was significantly altered in *cebpb* mutants, further validating the defected eosinophilopoiesis. (Results part, Page 9, Line 251-253)

3. Robustness of observations made on subsetting small datasets when there is only n=1 dataset. Figs 3E and 5D represent 4-category comparative maturation profiles of subpopulations of cells that amount to 0.3-3% of the total population of cells. However, it is easy to lose track of the actual n values for minor populations in complex scRNAseq datasets, which always appear large and data rich. The Methods section (line 453) suggests that there are approximately 11,000 cells/genotype (34K cells in total for 3 genotypes). Calculating therefore: 0.3% of 11K cells is 33

cells, and 16.2% of 33 cells (the smallest subset) is n=5 cells. Even though these categorical maturation profile data are presented in merely descriptive terms, and no analytical stats are performed, in any other setting, this would be considered too underpowered and under-replicated for the comparison to be of value. An ordered categorical comparative analysis could be performed (but with these low numbers per category, it is likely to be too underpowered to confirm that is a difference beyond what could be expected by chance alone).

Response: We're grateful for your valuable suggestion. In the revision, we triplicated the scRNA-Seq analysis (n=3) to strengthen our conclusions. Combined with 3 samples, the eosinophil numbers in the *cebpb* mutants increased to 105 cells, while still no Eb-P could be found in all *cebpb* mutant eosinophils (new Fig. 5D), demonstrating the impairment of eosinophil lineage commitment. The results also showed an increased tendency of pre.Eos and a decreased tendency of mat.Eos compared to WT subclusters (new Fig. 5D, Student's *t*-test), suggesting the developmental arrest at pre.Eos stage by *cebpb* deficiency. (Methods part, Page 17, Line 471-472; Results part, Page 9-10, Line 253-255)

4. Replication and validation of key functional experiments. Figure 7C bases the demonstrated of a shared repressive functional capacity for zebrafish *Cebp1* and human *C/EBP[epsilon]p27* on 7pprox.. n=15 embryos in these two test groups, with no indication of the degree of experimental replication. Indicate the degree of experimental replication. A reasonable expectation is that this experiment would be run at least n=3 times; n=15 embryos in each test group would be a reasonable minimum expectation the minimum given the variance seen here.

In this experiment, multiple *C/EBP* species had no biological effect, although their expression was validated by detecting EGFP expression from a carboxyl-fusion protein. The selection of *C/EBP[epsilon]p27* as the ortholog with exclusive shared functional capability relies on these being true negative results. For these no-effect species, what is the positive control that proves the carboxyl EGFP-fusion did not (or does not) inactivate the *C/EBP* transcription factor to which it was fused?

Response: Thank you for your advice. We have now clarified the repeat times and sample size information in the revised manuscript (Figure 7C, Line 987-988).

For your concern about the fusion protein activity, we performed luciferase assays to verify whether the fusion protein of *C/EBP*-eGFP affected the function of *C/EBP* factors (new Fig. S8, Methods part in Page 20, Line 544-552). Similar to Fig. 4E, the *DrCebp1*-eGFP could bind to the 3' UTR of the zebrafish *cebpb* gene and suppress the expression of *cebpb* (new Fig. S8A). We found that repressor isoforms *HsC/EBP[epsilon]^{p14}*-eGFP and *HsC/EBP[epsilon]^{p27}*-eGFP could significantly decrease human *MBP* promoter activity, while the activator isoforms *HsC/EBP[epsilon]^{p30}*-eGFP and *HsC/EBP[epsilon]^{p32}*-eGFP could induce *MBP* promoter activity (new Fig. S8B). *MmC/EBP[epsilon]*-eGFP could also induce mouse *MBP* promoter activity (new Fig. S8C). The above results proved that the fusion proteins didn't affect the activities of the *C/EBP* transcription factors. (Results part, Page 11, Line 299-302)

5. Line 350 acknowledges that'' The function of zebrafish eosinophils remains largely unknown.''

Li et al do reference several prior studies providing some evidence for shared functionalities between mammalian and zebrafish eosinophils. However the term “eosinophils” refers to a cell that has an affinity for eosin and other acidic stains in each species, not to a cells that even look alike and which have very different nuclei and granule morphologies. Assigning abundant expression of a MBP (major basic protein) related protein to this leukocyte lineage in zebrafish seems to me to be an important contribution from this report that supports the use of a common name that implies a common identity and function. I suggest the authors consider strengthening this point in their discussion.

Response: We're grateful for your valuable suggestions. We further discussed this point in the discussion part (Page 14, Line 373-375).

Minor points

1. Line 94-5. The insights for eosinophil-related diseases are potential future insights, not ones demonstrated in this paper (which did not model any specific disease), as is claimed here.

Response: Thank you for your correction. We've now modified this sentence and removed the descriptions related to diseases (Page 4, Line 94).

2. Lines 298-91 are not strictly a result – better a discussion point?

Response: Thank you for your correction. Since we had already written a similar sentence in the discussion part (Page 13, Line 349-351), we've now removed this redundant sentence in the result part.

3. Line 292. “functional” not “function”

Response: Thank you for your correction. We've fixed this careless mistake in our revised manuscript. (Page 11, Line 307).

For Reviewer #3 :

In this study, the authors elucidate the development of a key immune cell type, Eosinophils, in zebrafish. While some studies of adult zebrafish have characterized mature Eosinophils, the ontogeny of these cells is poorly understood. The authors apply a wide range of approaches to dissect the developmental trajectory of these cells and identify two key regulators, *cebpb1* and *cebpb*, which they demonstrate to be critical for eosinophil development. The authors go on to explore the conservation of these regulators in mice, and they identify both shared and disparate cell type-specific outcomes of gain and loss of function experiments. Overall, the authors perform a thorough characterization of eosinophil development in zebrafish and address the conservation of this process in mice. The integration of numerous experimental approaches is impressive, and the key points of each towards the main goal of the study provides a clear, logical through line. However, there are missing aspects of some important analyses that need to be addressed to facilitate proper interpretation of the data. And some information is lacking that would make the

study interesting and interpretable for a broader audience.

Major issues:

1. The authors surveyed adult tissue with the goal of describing eosinophil development, but it's not clear whether they expected to see cells at every developmental stage. Can the authors please clarify why they chose adult cells to explore the full developmental trajectory of these cells? Is there evidence that these cells are continuously developing over the lifespan of the animal?

Response: Thank you for your advice. In 7 dpf larvae, the percentage of eosinophils is quite low (~0.1%), thus it is difficult to obtain enough cells for scRNA-Seq through FACS (new Fig. S2A-B). In the kidney, eosinophils are relatively abundant (>1%) (new Fig. S2C-D). On the other hand, the kidney is the hematopoietic organ of zebrafish, comparative to the mammalian bone marrow, providing life-long definitive hematopoiesis from HSCs to various lineages. Therefore, we chose the cells in the adult kidney for scRNA-Seq and finally identified the eosinophil lineage from Eb-P to mat.Eos.

To validate whether eosinophils are continuously developing over the lifespan of zebrafish, we examined the overall eosinophil percentage of 7 dpf, 21 dpf, and 49 dpf zebrafish, and found that eosinophils could be detected in each stage (Fig. S2A-B). We also proved that eosinophils are widely distributed to adult organs such as the kidney, intestine, peritoneal fluid, spleen, and liver (Fig. S2C-D). According to the above results, we demonstrated that eosinophils are continuously developing over the lifespan in zebrafish (Methods part, Page 20, Line 554-563; Result part, Page 5-6, Line 127-141).

2. The authors should report additional quality control information for their single cell RNA-sequencing experiments, such as UMIs per cell type, quality control cutoffs, and numbers of genes expressed. Additionally, the methods section should be expanded to include information about the 10X kit version that was used, as well as the versions of all software packages (Seurat, Cellranger) that were used to analyze the data.

Response: Thank you for your correction. The quality control information was added (Page 17, Line 458-460 and 474-476). The versions of 10X kit (Page 16, Line 453) and all software packages (Page 16, Line 456, Page 17, Line 457 and 469; Page 18, Line 507-509; Page 19, Line 526-530) were also added in the revised manuscript.

3. From the current RNA velocity plot, it is difficult to visually summarize the directionality of the arrows. For instance, in the presumptive EoP's, it could be equally suggested by these data that the differentiation path is directly from EoP to mature eosinophil, rather than passing through the pre.Eos state. If the authors wish to support their Eosinophil trajectory inference using RNA velocity, it would be helpful to improve visibility of directionality in the velocity plot by adjusting the plotting parameters. Additionally, RNA velocity estimation relies heavily on the detection of spliced vs. unspliced mRNAs, and these can often vary from experiment to experiment or across different cell types. Can the authors please report these aspects of their input data in addition to an expanded methods section for any non-default parameters for the RNA velocity pipeline that they

used (or note default and the version if that was the case).

Response: We're grateful for your valuable suggestion. We modified the parameters ($\Delta T = 1$, $k_{Cells} = 40$) (Page 17, Line 469) to improve the visibility of directionality in the velocity plot (new Fig. 2F) as you suggested.

4. The population of "atypical erythrocytes" in the *cebpl*^{-/-} mutants is quite interesting, especially given the differences in zebrafish vs. mammalian erythrocytes (such as having a nucleus or not). To enhance the resource of this dataset and to provide additional information about this population of cells, it would be helpful for the authors to supply 1) an expanded list of cell type specific genes according to their annotations, and 2) a list of differentially expressed genes for each cell type for *cebpl*^{-/-} and *cebpb*^{-/-} fish compared to wildtype.

Response: We're grateful for your valuable suggestion. For the atypical cell population, we found the previous *eslec*⁺ "atypical erythrocytes" population (previous Fig 3C) was not repeatable when we triplicated the scRNAseq (new Fig 3C). Instead, another atypical cell population (~12%) with proliferative features was found in new *cebpl* mutant datasets (integrated from triplicates) (new Fig 3C and Fig. S5B). (Results part, Page 7, Line 191-192)

To enhance the resource of this dataset, we included the marker gene list of each cluster and the DEGs for each cell type for mutants (Supplementary Table 6). We also further subclustered eosinophils to list the relevant marker genes and DEGs (Supplementary Table 7). We also supplied the marker gene list of *eslec*⁺ KM cells (Fig. 2) as Supplementary Table 1, thereby completing all scRNA-Seq contents. (Methods part, Page 17, Line 467; Page 18, Line 490-492)

5. The authors note that adult zebrafish were collected and pooled for the scRNA-seq experiments. However, since eosinophils are an immune cell type that is responsive to environmental factors (such as infections), can the authors report the age of the animals and whether the genotypes that were compared came from the same clutch of siblings and collected when the animals were the same age? Additionally, if it was possible to sex the animals (i.e. if they were old enough), the authors should report whether females or males or a mixture were used for their experiments. All of this information is important to understand and interpret the RNA-seq data.

Response: We're grateful for your valuable suggestion. All adult zebrafish applied in the scRNA-Seq analysis were 3 months old. We didn't distinguish the genders of 5 animals in Fig. 2. For the newly performed scRNA-Seq of *cebpl* and *cebpb* mutants (new Fig. 3 and Fig. 5), we applied only females to try to avoid potential sex differences. For the newly performed bulk RNA-Seq assays, we applied larvae instead of adult fish, and their genders were unable to identify. We included the information to the method parts (Page 16, Line 448-449; Page 17, Line 471-472).

6. In Figure 7, the authors report that mouse BM cells were subjected to InfinityFlow analysis, but that method is not described anywhere in the manuscript. Especially given that UMAP dimensionality reduction is used to display the single cell RNA-seq data, the authors should add clarification to the text, methods, and figure to make it clear that a different data type is being displayed. The authors should additionally list the antibodies or panel that was used or include a

citation that lists them.

Response: Thank you for your question. We're sorry for our careless mistake in the legends of Fig. 7, actually we didn't perform InfinityFlow analysis in our research. It was a standard flow cytometry analysis of BM cells, using the in-built FlowJo UMAP plugin to perform the UMAP analysis. The antibodies were listed on Page 21, Line 583-587 and the citation was introduced on Page 21, Line 578. The related method descriptions were unchanged (Page 21, Line 577-588), while the wrong figure legends were corrected (Page 34, Line 951-952, 957).

7. The authors have made the data publicly available in GEO. However, for the scRNA-seq data, only the count matrix is available as a processed data file. To enable reproducible analysis, the authors should also upload their gene metadata and cell metadata (including cell type annotations). Additionally, deposition of the Seurat objects would be extremely helpful for enabling others to benefit from the large amount of data produced in their study (similarly to their re-analysis of the Tang, et al. data included in Supp Fig. 1).

Response: Thank you for your kind advice. We've uploaded the count matrix for each sample, while uploaded the gene and cell metadata for the integrated samples, as cell types were annotated after integration. The relevant Seurat RDS files were also submitted to GEO (GSE198314, token for reviewers: afsrqguwxpcfpin) as you suggested.

Minor issues:

1. To identify an eosinophil marker that is expressed early in development, the authors use a published single cell RNA-seq dataset of kidney cells. However, they 1) do not explain why the kidney is the proper place to look for these cells or 2) why the *gata2*⁺ cluster would be the eosinophils. Citations and explanation of this key background information should be added. (line 100-104)

Response: Thanks for your comments. The zebrafish kidney marrow is analogous to the mammalian bone marrow, where the HSCs give rise to various types of blood cells. We've added this background information to the results part (Page 4, Line 101-102). Besides, it was reported that the zebrafish eosinophils in KM are *gata2a*⁺, which was also cited in the results part (Page 4, Line 103, reference 21).

2. The authors indicate colocalization with *coro1a*⁺ suggests eosinophil identity, but they do not explain why. A citation or small explanation should be added here.

Response: Thanks for your comments. We've added the relevant citations to introduce that *coro1a* marks the entire myeloid cells (Page 5, Line 121-122), of which eosinophils are a subset.

3. "KM" eosinophils is an acronym that's used throughout the manuscript but never defined.

Response: Thank you for your correction. We've defined this acronym in our revised manuscript

(Page 4, Line 103).

4. There is a wide range of font sizes throughout the figures, including some that are illegible (i.e. axis labels in Fig 3C). These should be improved for readability.

Response: Thank you for your correction. The font sizes of the figures have been modified.

5. Numbers of cells should be noted wherever possible. For instance, no cell numbers are reported for the comparative analysis in Figure 3. Also, in line 453, it is not clear whether the ~34k cells is referring to the combination of the three genotypes or a subset of them.

Response: Thank you for your correction. We've indicated the total cell number (Page 17, Line 480-483) and the eosinophil counts (Page 17-18, Line 484-486) of each sample in the Methods part.

6. What version of the zebrafish genome and transcriptome were used to map the reads for all experiments? These should be added to the analysis portion of the methods.

Response: Thank you for your correction. In all the bioinformatics analysis in our research, the zebrafish genome version was always GRCz11. The related descriptions were added to our revised manuscript (Page 17, Line 457; Page 18, Line 507; Page 19, Line 526).

7. When the authors explain that they used RNA-seq in line 249, they should clarify that it is "bulk" RNA-seq to avoid confusion from the other single cell RNA-seq data in the paper and to highlight that this is a new experiment.

Response: Thank you for your kind advice. We've emphasized in all the bulk RNA-Seq sections that those data belong to "bulk" sequencing (Results part, Page 8, Line 217 and 219; Page 9, Line 227; Page 10, Line 265; and Methods part, Page 18, Line 494).

8. In line 251, the authors note that they're comparing RNA-seq data from adult, sorted eosinophils to ChIP-seq data from 7-dpf whole larvae. While I understand that this is likely due to experimental constraints on both the input requirements for ChIP-seq vs. RNA-seq as well as the challenge of using HS over-expression in adults, the authors should at least discuss this striking difference in age and tissue type and what implications it might have on their results.

Response: Thank you for your kind advice. We reformed bulk RNA-Seq (new Fig. 4 and 6) at the larval stage, which is in accordance with the stages of ChIP-Seq experiments (Fig. 4 and 6). (Methods part, Page 18, Line 496-500, and Results part, Page 8, Line 220; Page 10, Line 266)

Performing sequencing with the same cell population is not feasible in our hands, since sorted eosinophils could not reach enough cell amount for ChIP-Seq. Therefore, we utilized the whole larvae to perform ChIP-Seq assays alternatively.

9. For the UMAPs in figure 7, it would be very helpful to have direct labels for key cell

populations rather than just the legend. Additionally, axes (or a representation of them) should be added to the UMAP plots.

Response: Thank you for your advice. We've added the direct labels for "Eosinophils", "Neutrophils", and "Atypical Granulocytes" on the new Fig. 7A-B, and added the UMAP axes.

Typos:

1. Line 211-213 is confusing and should be clarified.

Response: Thank you for your correction. We've re-written this sentence to make it easier to understand (Page 9. Line 226-229).

2. There are a number of minor but distracting grammatical errors throughout the manuscript. An additional editing round to correct these would greatly improve overall readability.

Response: Thank you for your advice. As suggested, we tried to improve the English writing and correct the errors.

REVIEWERS' COMMENTS

Reviewer #1 (Remarks to the Author):

The authors have answered all my comments.

Reviewer #2 (Remarks to the Author):

I have reviewed the authors' responses to reviewer comments, focusing on those to the comments I made, and reviewed the revised manuscript. The authors have systematically considered all the suggestions.

1. Co-localisation.

Figs S1G and S3A directly address the point.

In the associated test (lines 120-123), ref 31 (Hall et al, 2007) is an unusual reference to link lyz+ expression with neutrophils. In this paper, Hall et al focused on lyz expression in a myelomonocytic population they predominantly described as macrophages, but acknowledged expression in other myeloid cells they considered to be probably granulocytes. It was several subsequent studies that definitively linked lyz+ expression predominantly with neutrophils (e.g. PMID: 17553562).

2. Augmented eosinophilopoiesis in the cebp1 mutants.

The appearance of a replacement population of only approximately the same proportion in a proportional analysis does not really address the point that a proportional change in %s needs to also be considered for what it means for changes in absolute numbers. However, this is a point in the rebuttal only. The HSPC-normalised data, and the analysis of adult kidney data address the point and support an increased "integrated eosinophil lineage" population.

3. Subsetting small datasets.

The increase in scRNAseq dataset size provided by the additional replicates strengthens the observation, and has provided a basis for analytical statistics to be applied.

4. Replication.

Although the comment in my first review might be seen to apply only to Fig 7, I had intended it apply throughout the paper, and I am pleased that another reviewer shared this concern. There has been systematic addition to figure legends of statements of the form "X independent experiments were conducted with $n \geq Y$ in each group) throughout the paper.

a. In all these instances, only one of the usually 3 replicates is shown, as the non-WT group generally has Y data points. Another way of looking at this information is that two-thirds of the data are not presented (2/3 when the number of experiments is 3). If the experiments have been done multiple times (usually 3), those data should be included in the paper or an explanation provide why they are not shown. The data could be presented as $n=3$ showing the 3 means of 3 independent experiments (if experiments are to be equally weighted), or by pooling data (if the individual in each replicate are to be equally weighted), or with the data from other experiments of which the feature one is representative included in a supplementary data file.

b. In the case of Fig 5A, the cebpb^{-/-} dataset contains only 13 points (whereas the legend says group sizes were ≥ 15) in each of 3 independent experiments.

c. The replication in Fig 7C is provided (the rebuttal line callout is incorrect),

The luciferase assays address the point about the functional integrity of the cebp fusion proteins.

Reviewer #3 (Remarks to the Author):

The authors have addressed a number of my comments through additional experiments, text modifications, expanded data availability, and clarification of points that were missing from the initial version of the manuscript. However, I have a few additional comments for revision:

1. The authors added helpful filtering metrics for their scRNA-seq processing, but QC of the resulting dataset(s) is still missing. Especially since new scRNA-seq experiments have been added, additional quantifications of umis/cell, number of genes/cell, proportions of each cell type across experiments are critical to evaluation of the data quality and consistency and are standard for reporting these types of data.

2. The RNA velocity plot in Figure 2 is somewhat improved, and it is helpful to have the specific parameters listed in the methods section, however the results (as currently visualized) still do not convincingly support the proposed trajectory. Updated visualizations available via the scVelo package could be a way to improve this. Additionally, the authors should report in the methods section, the % of intronic reads that they captured, which is helpful for interpreting data (this is available via the 10X web summary). I would like to note that RNA-velocity is only a somewhat useful tool that at best can validate known biology that is reflected in low-dimensional UMAP space, without a clear directional result in addition to fully orthogonal validation of the proposed trajectory direction (which is provided by responses to other reviewer comments), it is not a particularly valuable analysis.

3. The revision of Legend 7 is still too vague here: "BM cells of *Cebpe*^{+/+} and *Cebpe*^{-/-} mice were collected and applied to flow cytometry analysis." The UMAP visualization choice suggests dimensionality reduction from a high dimensional space - the authors should report the number of antibodies analyzed for this analysis in the legend to facilitate the interpretation of the plots.

Minor text comments:

- In line 472, change "applied" to "selected"
- In line 565, the method used here is technically RT-qPCR rather than Q-PCR. And if the authors want to just use the term Q-PCR, it should technically be qPCR but denoted somewhere what that is short for.

Reviewers point:

Reviewer #1 (Remarks to the Author):

The authors have answered all my comments.

Response: Thank you for your comment.

Reviewer #2 (Remarks to the Author):

I have reviewed the authors' responses to reviewer comments, focusing on those to the comments I made, and reviewed the revised manuscript. The authors have systematically considered all the suggestions.

1. Co-localisation.

Figs S1G and S3A directly address the point.

In the associated test (lines 120-123), ref 31 (Hall et al, 2007) is an unusual reference to link lyz+ expression with neutrophils. In this paper, Hall et al focused on lyz expression in a myelomonocytic population they predominantly described as macrophages, but acknowledged expression in other myeloid cells they considered to be probably granulocytes. It was several subsequent studies that definitively linked lyz+ expression predominantly with neutrophils (e.g. PMID: 17553562).

Response: Thank you for your kind suggestion. The previous ref 31 has been changed according to your suggestion. (new ref 31, Page 5, Line 124)

2. Augmented eosinophilopoiesis in the cebp1 mutants.

The appearance of a replacement population of only approximately the same proportion in a proportional analysis does not really address the point that a proportional change in %s needs to also be considered for what it means for changes in absolute numbers. However, this is a point in the rebuttal only. The HSPC-normalised data, and the analysis of adult kidney data address the point and support an increased "integrated eosinophil lineage" population.

Response: Thank you for your comments.

3. Subsetting small datasets.

The increase in scRNAseq dataset size provided by the additional replicates strengthens the observation, and has provided a basis for analytical statistics to be applied.

Response: Thank you for your comments.

4. Replication.

Although the comment in my first review might be seen to apply only to Fig 7, I had intended it apply throughout the paper, and I am pleased that another reviewer shared this concern. There has been systematic addition to figure legends of statements of the form "X independent experiments were conducted with $n \geq Y$ in each group) throughout the paper.

- a. In all these instances, only one of the usually 3 replicates is shown, as the non-WT group generally has Y data points. Another way of looking at this information is that two-thirds of the data are not presented (2/3 when the number of experiments is 3). If the experiments have been done multiple times (usually 3), those data should be included in the paper or an explanation provide why they are not shown. The data could be presented as n=3 showing the 3 means of 3 independent experiments (if experiments are to be equally weighted), or by pooling data (if the individual in each replicate are to be equally weighted), or with the data from other experiments of which the feature one is representative included in a supplementary data file.
- b. In the case of Fig 5A, the *cebpb*^{-/-} dataset contains only 13 points (whereas the legend says group sizes were ≥ 15) in each of 3 independent experiments.
- c. The replication in Fig 7C is provided (the rebuttal line callout is incorrect),

Response: Thank you for your suggestions.

For point a, we will accordingly provide the data from repeat experiments in the following data submission.

For point b, thank you for pointing out our mistake. The legends of Fig. 5a and Fig. 5b were written oppositely in the last version. In fact, the legend of Fig. 5b should be Fig. 5a, which indicates the group sizes are ≥ 10 . The order of legends has been corrected in the revision, and the precise n values are declared (Page 33, Line 933). We are sorry for the mistake. Now we have corrected it accordingly.

For point c, we are sorry for the mislabeling in last rebuttal.

The luciferase assays address the point about the functional integrity of the *cebpb* fusion proteins.

Response: Thank you for your comments.

Reviewer #3 (Remarks to the Author):

The authors have addressed a number of my comments through additional experiments, text modifications, expanded data availability, and clarification of points that were missing from the initial version of the manuscript. However, I have a few additional comments for revision:

1. The authors added helpful filtering metrics for their scRNA-seq processing, but QC of the resulting dataset(s) is still missing. Especially since new scRNA-seq experiments have been added, additional quantifications of umis/cell, number of genes/cell, proportions of each cell type across experiments are critical to evaluation of the data quality and consistency and are standard for reporting these types of data.

Response: Thank you for your suggestions. The QC information for umis/cell (nCount) and genes/cell (nFeature) is provided in the Methods (Page 17, Line 460-461 and Line 478-479), while the proportions of cell types are listed in the supplementary information.

2. The RNA velocity plot in Figure 2 is somewhat improved, and it is helpful to have the specific

parameters listed in the methods section, however the results (as currently visualized) still do not convincingly support the proposed trajectory. Updated visualizations available via the scVelo package could be a way to improve this. Additionally, the authors should report in the methods section, the % of intronic reads that they captured, which is helpful for interpreting data (this is available via the 10X web summary). I would like to note that RNA-velocity is only a somewhat useful tool that at best can validate known biology that is reflected in low-dimensional UMAP space, without a clear directional result in addition to fully orthogonal validation of the proposed trajectory direction (which is provided by responses to other reviewer comments), it is not a particularly valuable analysis.

Response: Thank you for your suggestions. Accordingly, the percentage of intronic reads is provided in the revised manuscript (Page 17, Line 471).

3. The revision of Legend 7 is still too vague here: “BM cells of *Cebpe*^{+/+} and *Cebpe*^{-/-} mice were collected and applied to flow cytometry analysis.” The UMAP visualization choice suggests dimensionality reduction from a high dimensional space - the authors should report the number of antibodies analyzed for this analysis in the legend to facilitate the interpretation of the plots.

Response: Thank you for your suggestions. The number of antibodies has been declared in the legends (Page 34, Line 962-963 and Line 968).

Minor text comments:

- In line 472, change “applied” to “selected”
- In line 565, the method used here is technically RT-qPCR rather than Q-PCR. And if the authors want to just use the term Q-PCR, it should technically be qPCR but denoted somewhere what that is short for.

Response: Thank you for your suggestions. The relevant descriptions have been modified (Page 17, Line 475; Page 20, Line 569 and Line 573).